# Machine Unlearning under Retain–Forget Entanglement

**Jingpu Cheng**
Department of Mathematics
National University of Singapore
`chengjingpu@u.nus.edu`

**Ping Liu**
Department of Computer Science
University of Nevada Reno
`pino.pingliu@gmail.com`

**Qianxiao Li**
Department of Mathematics
Institute for Functional Intelligent Materials
National University of Singapore
`qianxiao@nus.edu.sg`

**Chi Zhang**[*]
Department of Mathematics
National University of Singapore
`czhang24@nus.edu.sg`

## ABSTRACT

Forgetting a subset in machine unlearning is rarely an isolated task. Often, retained samples that are closely related to the forget set can be unintentionally affected, particularly when they share correlated features from pretraining or exhibit strong semantic similarities. To address this challenge, we propose a novel two-phase optimization framework specifically designed to handle such retain–forget entanglements. In the first phase, an augmented Lagrangian method increases the loss on the forget set while preserving accuracy on less-related retained samples. The second phase applies a gradient projection step, regularized by the Wasserstein-2 distance, to mitigate performance degradation on semantically related retained samples without compromising the unlearning objective. We validate our approach through comprehensive experiments on multiple unlearning tasks, standard benchmark datasets, and diverse neural architectures, demonstrating that it achieves effective and reliable unlearning while outperforming existing baselines in both accuracy retention and removal fidelity. Our code is available here.

## 1 INTRODUCTION

The indelible memory of machine learning systems presents a paradoxical challenge: what happens when we need algorithms to forget? Consider a face recognition system deployed for secure access. When an employee resigns, their biometric signature cannot simply be deactivated—it must be completely expunged from the underlying machine learning model. This ability to erase specific information extends to the broader concept known as "machine unlearning" (Cao and Yang, 2015), which aims to selectively remove the impact of specific data from trained models. In fact, the importance of unlearning extends beyond the general concept itself, with critical applications in meeting legal obligations (Mantelero, 2013), mitigating harmful representational biases (Mehrabi et al., 2021), and repairing models from mislabeled or poisoned training data (Northcutt et al., 2021).

Recent work has investigated a range of unlearning scenarios, including random-sample unlearning (Golatkar et al., 2020a; Izzo et al., 2021), class-wise unlearning (Kurmanji et al., 2023), and concept-level unlearning, where the forget set does not necessarily align with class labels (Zhu et al., 2024). More recently, several methods (Foster et al., 2024; Seo et al., 2025; Xu et al., 2024) have introduced efficient post-hoc or feature-space–aware solutions. Together, these approaches have significantly advanced our understanding of what it means for a model to "forget".

Yet, forgetting is rarely an isolated task. Removing the influence of one group of data often directly affects another group that is closely correlated with it. For instance, forgetting toxic statements involving a minority group may inadvertently alter the model's behavior on non-toxic statements

---

[*]Corresponding: `czhang24@nus.edu.sg`

about the same group (Shen et al., 2024). Similarly, forgetting one subclass of images within a broader category can disrupt predictions on closely related subclasses (Fan et al., 2024a). Existing works typically assess retain performance by averaging over the entire retain set, paying little attention to these sensitive, correlated subsets, where performance is both more fragile and more consequential.

We therefore focus on the challenge of *retain–forget entanglement*, where certain retained samples are closely tied to the forget set and particularly susceptible to unintended degradation. To mitigate the resulting performance drops in these sensitive subsets, we propose a two-stage framework based on constrained optimization. In the first stage, an augmented Lagrangian method enforces forgetting by increasing the loss on the forget set while preserving accuracy on less-correlated retained samples. In the second stage, the model is refined through gradient projection to restore performance on retained samples that are more strongly correlated with the forget set, without compromising the forgetting objective. To further stabilize the process and enhance generalization, we also regularize the loss distribution using the Wasserstein-2 distance during this stage.

We evaluate our method across a variety of subclass-level unlearning scenarios, covering diverse forgetting tasks, multiple neural network architectures, and standard benchmark datasets. The results demonstrate that our approach consistently achieves effective forgetting while maintaining high accuracy on the retained data. In structured selective unlearning settings, it significantly outperforms prior methods, demonstrating robustness and reliability without compromising the intended forgetting effect. Importantly, it preserves performance on retained samples that are closely related to the forget set, ensuring that sensitive subsets remain largely unaffected.

Our contributions can be summarized as follows:

- **Highlighting retain–forget entanglement:** We focus on a correlation-aware unlearning setting, where the forget set is entangled with another group of data. This setting better reflects real-world unlearning demands and introduces new technical challenges due to significant distributional overlap with the retained data.
- **A novel two-stage unlearning framework:** We propose a two-stage optimization-based framework to address this challenge. The first stage uses an augmented Lagrangian method to enforce forgetting while preserving performance on less-correlated samples. The second stage applies gradient projection with Wasserstein-2 distance regularization to recover performance on sensitive retained samples without compromising the forgetting objective.
- **Comprehensive evaluation:** We provide a comprehensive empirical evaluation across diverse tasks, architectures, and datasets, demonstrating that our method achieves strong forgetting performance while retaining accuracy on preserved data.

## 2 RELATED WORKS

**Constrained Optimization in Machine Learning**   Constrained optimization is widely used in machine learning to enforce domain-specific requirements like fairness and safety (Cotter et al., 2019; Zafar et al., 2019; Achiam et al., 2017; Liu et al., 2022). In fairness-aware learning, these constraints prevent discriminatory predictions and are naturally framed as optimization problems (Donini et al., 2018; Zafar et al., 2019; Caton and Haas, 2024). Classical techniques, such as penalty methods Berk et al. (2017) and Lagrangian-based approaches (Cruz et al.; Celis et al., 2019; Cotter et al., 2019; Lokhande et al., 2020), have proven effective in these settings. Similarly, in reinforcement learning, safety constraints guide agents away from risky actions (Chow et al., 2018; Liu et al., 2022), often handled through primal-dual optimization to penalize constraint violations (Achiam et al., 2017; Liang et al., 2018; Bohez et al., 2019).

**Machine Unlearning**   The concept of machine unlearning was formalized by (Cao and Yang, 2015), requiring model outputs indistinguishable from retraining without the deleted data. However, full retraining is often infeasible for large-scale models, motivating approximate methods (Golatkar et al., 2020a;b; Izzo et al., 2021; Thudi et al., 2022; Mehta et al., 2022). Many build on the framework of Ginart et al. (2019), gradient-based updates (Golatkar et al., 2020a; Fan et al., 2024b; Patel and Qiu, 2025), sparsity-based pruning (Jia et al., 2023), prompt editing (Liu et al., 2024), fisher and influence based methods (Foster et al., 2024; Shi et al., 2024; Wu et al., 2022) and adversarial approaches (Di et al., 2024). In particular, fine-tuning (Warnecke et al., 2021; Zhang et al., 2024; 2025) has been shown to be an effective post-training approach and is widely used in machine unlearning. For example, Kurmanji et al. (2023) proposed post-training methods by discouraging the model from

correctly predicting labels on a forget set. Meanwhile, retain–forget entanglement has been show to impact unlearning, where accuracy drops are often concentrated on retained examples most similar to the forget set (Zhao et al., 2024; Chang and Lee, 2025). For example, subclass-level forgetting (Zhu et al., 2024; Foster et al., 2024; Seo et al., 2025) considers settings where the forget subclass is semantically close to other subclasses. Yet performance is often reported as an average over the entire retain set, which can mask degradation on the correlated subset. Similarly, in LLM unlearning (Maini et al., 2024; Jin et al., 2024; Chang and Lee, 2025; Choi et al., 2025), a neighbor set is often used for evaluation or regularization; in concept erasure for generative models (Gandikota et al., 2024; Xie et al., 2025; Liu and Zhang, 2025), preservation sets are used to maintain overall performance. Yet, these sets need not be closely related to the erased targets, whereas we explicitly decompose the retain set into adjacent and remote subsets and report the performance on both of them.

## 3 PROBLEM FORMULATION

Let $\mathcal{D} = \{(x_i, y_i)\}_{i=1}^N$ be a dataset of $N$ samples, where $x_i \subset \mathcal{X}$ denotes an input and $y_i \in \mathcal{Y}$ is its corresponding label. Let $f_{\theta_0}(x)$ be a model trained on $\mathcal{D}$ with parameters $\theta_0$. Given a subset $\mathcal{D}_f \subset \mathcal{D}$, the goal of machine unlearning is to obtain updated parameters $\tilde{\theta}$ such that the resulting model $f_{\tilde{\theta}}(x)$ effectively forgets $\mathcal{D}_f$, while preserving performance on the remaining data $\mathcal{D}_r := \mathcal{D} \setminus \mathcal{D}_f$.

Classical formulations of machine unlearning typically do not assume further structures in the retain dataset. However, in many applications, forgetting $\mathcal{D}_f$ affects not only average performance on $\mathcal{D}_r$, but disproportionately impacts a correlated portion inside $\mathcal{D}_r$ (Fan et al., 2024a). We therefore conceptually split the retain set into two parts:

$$\mathcal{D}_r = \mathcal{D}_r^{\text{adj}} \cup \mathcal{D}_r^{\text{rem}}, \qquad \mathcal{D}_r^{\text{adj}} \cap \mathcal{D}_r^{\text{rem}} = \emptyset.$$

Here, the adjacent retain set $\mathcal{D}_r^{\text{adj}}$ consists of retained examples that are correlated with $\mathcal{D}_f$ and thus more sensitive to forgetting, while the remote retain set $\mathcal{D}_r^{\text{rem}}$ comprises the remaining, less-related retained examples, which we refer to as the remote samples.

In practice, the entanglement between the forget set and retained samples can arise from different sources. One common scenario is subclass-level unlearning, where the forget set constitutes a fine-grained subclass within a broader class. For example, if a model is trained on the 20 superclasses of CIFAR-100 and the forget set consists of one subclass, we can define $\mathcal{D}_r^{\text{adj}}$ as the remaining samples from the same superclass and $\mathcal{D}_r^{\text{rem}}$ as the rest of the dataset. Another scenario occurs when retained samples form a semantically related group with the forget set. For instance, in a language dataset containing normal and offensive sentences, comments referring to the same group of people may be strongly correlated with the forget set.

The goals of this retain-forget entangled machine unlearning are therefore to obtain an updated model such that

1. The model retains its performance on $\mathcal{D}_r$, especially on samples belonging to $D_r^{\text{adj}}$ that have strong correlation to $\mathcal{D}_f$.
2. The model forgets the forget set $\mathcal{D}_f$ by removing or mitigating its influence.

It is important to note that the definition of "forgetting" can vary depending on the application. In privacy-focused contexts (Cao and Yang, 2015), the objective is often for $f_{\tilde{\theta}}$ to emulate a model retrained from scratch on the retained set $\mathcal{D}_r$. In contrast, there are scenarios that prioritize maximally reducing the model's performance on the forget set $\mathcal{D}_f$, as studied in (Choi and Na, 2023). This approach is particularly relevant when $\mathcal{D}_f$ contains undesirable patterns, such as social biases, offensive content, or behaviors subject to withdrawal requests, where the goal is for the model to completely disregard the influence of these samples. In this work, we adopt the latter perspective.

## 4 METHODS

Machine unlearning naturally poses a multi-objective challenge: removing the influence of the forget set while maintaining overall performance. In the retain-forget entangled setting, this becomes more difficult due to the semantic and distributional entanglement between the forget set $\mathcal{D}_f$ and the

strongly correlated retain set $\mathcal{D}_r^{\text{adj}}$. To address this challenge, we introduce a two-stage optimization framework in this section.

## 4.1 STAGE 1: FORGETTING VIA CONTROLLED OPTIMIZATION

The first stage of our framework aims to aggressively increase the loss on the forget set while preventing substantial degradation on the less-related retain set. Formally, let $\mathcal{L}_f(\theta) := \mathcal{L}_f(\theta; \mathcal{D}_f)$, $\mathcal{L}_r^{\text{adj}}(\theta) := \mathcal{L}_r^{\text{adj}}(\theta; \mathcal{D}_r^{\text{adj}})$, and $\mathcal{L}_r^{\text{rem}}(\theta) := \mathcal{L}_r^{\text{rem}}(\theta; \mathcal{D}_r^{\text{rem}})$ denote the losses on $\mathcal{D}_f$, $\mathcal{D}_r^{\text{adj}}$, and $\mathcal{D}_r^{\text{rem}}$, and let $\theta_0$ be the parameters of the original model. We formulate Stage 1 as the constrained optimization problem

$$\min_\theta \ -\mathcal{L}_f(\theta) \quad \text{subject to} \quad \mathcal{L}_r^{\text{rem}}(\theta) = \mathcal{L}_r^{\text{rem}}(\theta_0). \tag{1}$$

We adopt an augmented Lagrangian formulation (Bertsekas, 2014) to provide an adaptive way of balancing the objective and the constraint:

$$\mathcal{L}_{\text{aug}}(\theta; \lambda, \mu) = -\mathcal{L}_f(\theta) + \lambda\big(\mathcal{L}_r^{\text{rem}}(\theta) - \mathcal{L}_r^{\text{rem}}(\theta_0)\big) + \frac{\mu}{2}\big(\mathcal{L}_r^{\text{rem}}(\theta) - \mathcal{L}_r^{\text{rem}}(\theta_0)\big)^2, \tag{2}$$

where $\lambda$ is the Lagrange multiplier and $\mu > 0$ is a penalty coefficient. We initialize $\lambda = 0$ and iteratively update $\theta$ via gradient descent,

$$\theta \leftarrow \theta - \eta \nabla_\theta \mathcal{L}_{\text{aug}}(\theta; \lambda, \mu), \tag{3}$$

followed by updating the multiplier according to constraint violation:

$$\lambda \leftarrow \lambda + \mu\big(\mathcal{L}_r^{\text{rem}}(\theta) - \mathcal{L}_r^{\text{rem}}(\theta_0)\big). \tag{4}$$

This iterative update scheme adaptively tightens or relaxes the penalty as needed, avoiding the need to manually tune a fixed trade-off coefficient. The objective of Stage 1 is to enforce unlearning on the forget set $\mathcal{D}_f$ while preserving performance on the less-related retained subset $\mathcal{D}_r^{\text{rem}}$.

## 4.2 STAGE 2: $W_2$-DISTANCE GUIDED PROJECTED GRADIENT DESCENT (W-PGD)

Importantly, we refrain from explicitly optimizing over the strongly correlated retain set $\mathcal{D}_r^{\text{adj}}$ in Eq (2) to avoid conflicting gradients (see Appendix B.5). As a result, the model achieves low accuracy on the forget set $\mathcal{D}_f$ while maintaining strong performance on the remote retain set $\mathcal{D}_r^{\text{rem}}$. However, due to the semantic or distributional overlap between $\mathcal{D}_f$ and $\mathcal{D}_r^{\text{adj}}$, performance on the adjacent retain set $\mathcal{D}_r^{\text{adj}}$ typically degrades. The objective of the second stage is to restore the model's accuracy on $\mathcal{D}_r^{\text{adj}}$ while preserving the performance on $\mathcal{D}_f$ and $\mathcal{D}_r^{\text{rem}}$.

### 4.2.1 IS CLASSICAL PROJECTED GRADIENT DESCENT GOOD ENOUGH?

We begin by aiming to improve the performance on the adjacent retain set using the classical Projected Gradient Descent (PGD) framework (Bertsekas, 1999), but adopt its first-order (linearized) projection

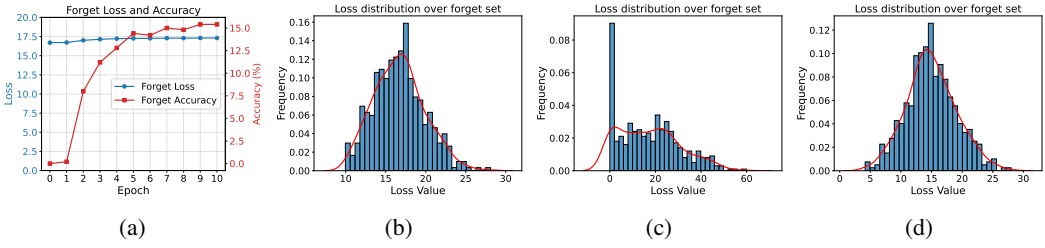

| (a) | (b) | (c) | (d) |

Figure 1: Training dynamics of PGD and cross-entropy loss distributions on $\mathcal{D}_f$. (a) Loss and accuracy curves of PGD during the second stage; (b) Original loss distribution on $\mathcal{D}_f$ after the first stage; (c) Loss distribution on $\mathcal{D}_f$ after applying PGD in the second stage; (d) Loss distribution on $\mathcal{D}_f$ after applying W-PGD. Comparing figure (b) and (c), PGD notably **skews the loss distribution**, with some samples attaining near-zero loss. In contrast, W-PGD (d) preserves a distribution closer to the original and effectively avoids assigning low loss to forget set samples.

variant, as widely used in multi-task learning (Yu et al., 2020; Farajtabar et al., 2020). In this approach, the update modifies the gradient of $\mathcal{L}_r^{\text{adj}}$ by removing its components aligned with the gradients of $\mathcal{L}_f$ and $\mathcal{L}_r^{\text{rem}}$:

$$\theta \leftarrow \theta - \eta \left( \nabla_\theta \mathcal{L}_r^{\text{adj}} - \text{Proj}_V \nabla_\theta \mathcal{L}_r^{\text{adj}} \right), \quad \text{where } V = \text{span} \left\{ \nabla_\theta \mathcal{L}_f, \nabla_\theta \mathcal{L}_r^{\text{rem}} \right\}, \tag{5}$$

Yet, this conventional optimization technique can exhibit significant performance degradation when applied to correlation-aware machine unlearning. As illustrated in Figure 1a, although the average loss on the forget set $\mathcal{D}_f$ (blue line) remains stable under PGD, the prediction accuracy (red line) on $\mathcal{D}_f$ increases steadily. This counterintuitive behavior stems from the strong semantic and distributional entanglement between $\mathcal{D}_f$ and the adjacent retained set $\mathcal{D}_r^{\text{adj}}$: minimizing loss on the latter inadvertently reduces the loss on similar samples in $\mathcal{D}_f$. To compensate and maintain the mean loss on $\mathcal{D}_f$, the model disproportionately increases the loss on less similar samples, resulting in a polarized loss distribution, as depicted in Figure 1c. This observation exposes a critical limitation of standard PGD: it lacks the ability to control accuracy-level changes when loss redistribution is uneven. Indeed, preserving only the mean loss provides no guarantees on the proportion of samples with low loss values, often resulting in *high accuracy* on the forget set as many samples remain correctly predicted.

### 4.2.2 Gradient Projection with Wasserstein Distance Regularization

The failure of gradient projection using mean losses motivates the need for a more fine-grained control over the forgetting behavior. To this end, we propose to explicitly regularize the distributional shift in loss values on $\mathcal{D}_f$ by incorporating a Wasserstein-2 distance penalty.

The Wasserstein-2 distance, denoted $W_2$, is a principled metric for comparing probability distributions (Vaserstein, 1969). Given two probability distributions $P$ and $Q$ over $\mathbb{R}^d$, the $W_2$ distance is defined as

$$W_2(P, Q) = \left( \inf_{\gamma \in \Gamma(P,Q)} \int_{\mathbb{R}^d \times \mathbb{R}^d} \|u - v\|^2 \, d\gamma(u, v) \right)^{1/2}, \tag{6}$$

where $\Gamma(P, Q)$ denotes the set of joint distributions with marginals $P$ and $Q$. In our setting, $P$ and $Q$ represent empirical distributions of scalar loss values, admitting a closed-form expression for the $W_2$ distance. Specifically, given two collections of loss values $\{a_1, \ldots, a_N\}$ and $\{b_1, \ldots, b_N\}$, the corresponding empirical distributions are defined as $P = \frac{1}{N} \sum_i \delta_{a_i}$ and $Q = \frac{1}{N} \sum_i \delta_{b_i}$, where $\delta$ denotes the Dirac delta function. Then, after sorting the samples as $\bar{a}_1 \leq \cdots \bar{a}_N$ and $\bar{b}_1 \leq \cdots \bar{b}_N$, the Wasserstein-2 distance is simply given by

$$W_2(P, Q) = \left( \frac{1}{N} \sum_{i=1}^N (\bar{a}_i - \bar{b}_i)^2 \right)^{1/2}. \tag{7}$$

We define the empirical loss distribution over the forget set under parameters $\theta$ as

$$P_\theta^{\text{forget}} := \frac{1}{|\mathcal{D}_f|} \sum_{(x_i, y_i) \in \mathcal{D}_f} \delta_{\ell(f_\theta(x_i), y_i)}, \tag{8}$$

where $\ell$ denotes the cross-entropy loss. Let $\bar{\theta}$ denote the model parameters after the first stage. To constrain the mean and distributional shape of the loss over $\mathcal{D}_f$, we define a modified loss function:

$$\tilde{\mathcal{L}}_f(\theta) := (1 - \alpha)\mathcal{L}_f(\theta) + \alpha W_2^2 \left( P_{\bar{\theta}}^{\text{forget}}, P_\theta^{\text{forget}} \right), \tag{9}$$

where $\alpha \in [0, 1]$ is a hyperparameter balancing the influence of the mean and distributional components. We then modify the gradient projection update to project the gradient of $\mathcal{L}_r^{\text{adj}}$ onto the orthogonal complement of the space spanned by the gradients of $\tilde{\mathcal{L}}_f$ and $\mathcal{L}_r^{\text{rem}}$:

$$\theta \leftarrow \theta - \eta \left( \nabla_\theta \mathcal{L}_r^{\text{adj}}(\theta) - \text{Proj}_V \nabla_\theta \mathcal{L}_r^{\text{adj}}(\theta) \right), \quad \text{where } V = \text{span} \left\{ \nabla_\theta \tilde{\mathcal{L}}_f(\theta), \nabla_\theta \mathcal{L}_r^{\text{rem}}(\theta) \right\}. \tag{10}$$

This modified gradient projection method (W-PGD) enforces $\tilde{\mathcal{L}}_f(\theta)$ to be mostly unchanged during the update, while allowing the model to recover performance on the adjacent retain set $\mathcal{D}_r^{\text{adj}}$, as indicated by the following proposition.

---

**Algorithm 1** Two-Stage Machine Unlearning

---

**Input:** Forget set $\mathcal{D}_f$, retain sets $\mathcal{D}_r^{\text{adj}}$ and $\mathcal{D}_r^{\text{rem}}$, learning rates $\eta_1, \eta_2$, penalty coefficient $\mu$ and $\alpha$, number of iterations $K, M$. Initialize $\theta = \theta_0$, $\lambda = 0$, compute $\mathcal{L}_r^{\text{rem}}(\theta_0)$
**Stage 1: Augmented Lagrangian optimization**
**for** $i = 1$ **to** $K$ **do**
     Compute $\mathcal{L}_{\text{aug}}(\theta; \lambda, \mu) = -\mathcal{L}_f(\theta) + \lambda \left(\mathcal{L}_r^{\text{rem}}(\theta) - \mathcal{L}_r^{\text{rem}}(\theta_0)\right) + \frac{\mu}{2} \left(\mathcal{L}_r^{\text{rem}}(\theta) - \mathcal{L}_r^{\text{rem}}(\theta_0)\right)^2$
     Update $\theta$: $\theta \leftarrow \theta - \eta_1 \nabla_\theta \mathcal{L}_{\text{aug}}(\theta; \lambda, \mu)$
     Update $\lambda$: $\lambda \leftarrow \lambda + \mu \left(\mathcal{L}_r^{\text{rem}}(\theta) - \mathcal{L}_r^{\text{rem}}(\theta_0)\right)$
**end for**
**Stage 2: $W_2$-distance guided gradient projection optimization**
**for** $j = 1$ **to** $M$ **do**
     Compute $\tilde{\mathcal{L}}_f(\theta) = (1 - \alpha)\mathcal{L}_f(\theta) + \alpha W_2^2 \left(P_{\bar{\theta}}^{\text{forget}}, P_\theta^{\text{forget}}\right)$
     Compute $\nabla_\theta \tilde{\mathcal{L}}_f, \nabla_\theta \mathcal{L}_r^{\text{adj}}, \nabla_\theta \mathcal{L}_r^{\text{rem}}$
     Update: $\theta \leftarrow \theta - \eta_2 \left(\nabla_\theta \mathcal{L}_r^{\text{adj}} - \text{Proj}_V \nabla_\theta \mathcal{L}_r^{\text{adj}}\right)$, where $V = \text{span}\left\{\nabla_\theta \tilde{\mathcal{L}}_f, \nabla_\theta \mathcal{L}_r^{\text{rem}}\right\}$
**end for**
**Output:** Unlearned model parameters $\theta$

---

**Proposition 4.1.** *Assume* $\tilde{\mathcal{L}}_f(\theta)$, $\mathcal{L}_r^{adj}(\theta)$ *and* $\mathcal{L}_r^{rem}(\theta)$ *are twice continuously differentiable to* $\theta$. *Let* $\Delta\theta$ *be the update of* $\theta$ *introduced by* (10). *Then, for sufficiently small* $\eta > 0$, *we have:*

(i) *The change in* $\tilde{\mathcal{L}}_f$ *and* $\mathcal{L}_r^{rem}$ *is at most second order in* $\eta$, *i.e.*

$$\tilde{\mathcal{L}}_f(\theta + \Delta\theta) - \tilde{\mathcal{L}}_f(\theta) = O(\eta^2), \quad L_r^{rem}(\theta + \Delta\theta) - L_r^{rem}(\theta) = O(\eta^2).$$

(ii) *If* $\nabla_\theta \mathcal{L}_r^{adj}(\theta)$ *is not in the span of* $\nabla_\theta \tilde{\mathcal{L}}_f(\theta)$ *and* $\nabla_\theta \mathcal{L}_r^{rem}(\theta)$, *then*

$$\mathcal{L}_r^{adj}(\theta + \Delta\theta) - \mathcal{L}_r^{adj}(\theta) = -c\,\eta + O(\eta^2),$$

*for some positive* $c$ *(depending on* $\theta$*). Hence, for sufficiently small* $\eta$, $\mathcal{L}_r^{adj}$ *strictly decreases.*

Moreover, compared to the projected gradient descent where no guarantee on the accuracy of the forget set is provided, the following proposition provides a bound on the accuracy of the forget set after the update. Specifically, let $n$ be the number of superclasses in the classification task, and $\text{Acc}_f(\theta)$ denote the accuracy of the model on the forget set $\mathcal{D}_f$. We have:

**Proposition 4.2.** *Let* $m > \log n$ *and* $\varepsilon > 0$. *Suppose that* $|\tilde{\mathcal{L}}_f(\theta) - \tilde{\mathcal{L}}_f(\bar{\theta})| < \varepsilon$, *and* $\ell(f_{\bar{\theta}}(x_i), y_i) \geq m$ *for all* $(x_i, y_i) \in \mathcal{D}_f$. *Then, the accuracy on* $\mathcal{D}_f$ *is upper bounded by:*

$$\text{Acc}_f(\theta) \leq \frac{1}{(m - \log n)^2} \left(\frac{1 - \alpha}{\alpha} + \sqrt{\frac{\varepsilon}{\alpha}}\right)^2. \tag{11}$$

The proposition indicates that when the minimum loss for model with parameter $\bar{\theta}$ is large and $\alpha$ is above zero, the accuracy of the forget set after W-PGD is bounded by a small constant, ensuring the forgetting behavior of the model. Notice that for a given $\varepsilon$, the upper bound in the above proposition is minimized when $\alpha = 1$, i.e., when the Wasserstein distance is fully utilized. However, in practice where we assess each loss value in mini-batch sense, we observe that choosing $\alpha = 1$ may not achieve the best overall performance (see Ablation studies on $\alpha$ in Appendix B.5). In our following experiments, we set $\alpha$ to be $0.5$. As shown in Figure 1d, the loss distribution of the forget set after W-PGD is more uniform compared to that of PGD, maintaining zero accuracy on the forget set.
In summary, the complete two-stage unlearning procedure is presented in Algorithm 1. We evaluate our method in correlation-aware unlearning scenarios across multiple datasets and architectures.

### 4.3 DISCUSSIONS ON THE TWO STAGES

The goal of the first stage is relatively straightforward, as the disentanglement between the forget set and the remote retain set makes the task less challenging. While alternative approaches, such as

adding fixed-weight penalty terms, could in principle achieve a similar trade-off with carefully tuned hyperparameters, the augmented Lagrangian formulation offers a key advantage: it introduces an adaptive multiplier that automatically balances the objective and constraint terms throughout training, resulting in a process that is more stable and less sensitive to hyperparameter choices.

A key component in the second stage is the use of distributional constraints formulated via $W_2$ distances. Prior work such as Golatkar et al. (2020a) also enforces the distributional constraints by estimating KL divergence between parameter distributions under a Gaussian prior. As comparison, $W_2$ admits a *closed-form solution* for one-dimensional empirical distributions via sorting, whereas KL divergence generally requires *density estimation* or *strong parametric assumptions*, introducing approximation errors and additional computational cost (Lv et al., 2024). Therefore, the use of $W_2$ distances makes computation far more convenient, avoiding the approximations (e.g., kernel estimation) or prior assumptions typically needed for KL divergence, while still providing a principled and effective distributional constraint.

## 5 EXPERIMENTS

In this section, we conduct a comprehensive evaluation of our proposed method across various machine unlearning scenarios. To ensure the generality of our findings, we design experiments that span multiple unlearning tasks, various benchmark datasets, and different network architectures.

### 5.1 EXPERIMENTAL SETUPS

**Datasets:** Following prior work on machine unlearning (Kurmanji et al., 2023), we conduct experiments on CIFAR-100 (Krizhevsky et al., 2009), TinyImageNet (Le and Yang, 2015), and a safety-critical language task on ToxiGen (Hartvigsen et al., 2022). For the vision benchmarks, we adopt the superclass organization (CIFAR-100: 20 superclasses × 5 subclasses; TinyImageNet: 10 semantic groups; see Appendix B.1). Given a selected forget subset $\mathcal{D}_f$ (a labeled subclass within a superclass), we define $\mathcal{D}_r^{\text{adj}}$ as the remaining samples from the same superclass and $\mathcal{D}_r^{\text{rem}}$ as all other retained samples. For the language task, we use ToxiGen with a normal/toxic binary classifier, where $\mathcal{D}_f$ consists of toxic sentences about the LGBTQ group, $\mathcal{D}_r^{\text{adj}}$ contains non-toxic sentences about the same group, and $\mathcal{D}_r^{\text{rem}}$ includes other sentences. This setup instantiates an unlearning scenario with retain-forget entanglement, where $\mathcal{D}_r^{\text{adj}}$ forms a semantic subgroup closely related to the forget set.

**Baseline methods:** We compare our approach against various unlearning methods, including: **Gradient Ascent (GA)** (Thudi et al., 2022): Train the model by maximizing the loss on the forget set. **Fine-Tune (FT)** (Warnecke et al., 2021; Golatkar et al., 2020a): Fine-tune the model on the retained set. **SCRUB** (Kurmanji et al., 2023): perform gradient ascent on the forget set and descent on the retain set simultaneously with distillation from the original model. $\ell_1$**-sparse** (Jia et al., 2023): fine-tune the model on the retain set with $\ell_1$-norm regularization on the model. **SSD** (Selective Synaptic Dampening) (Foster et al., 2024): post-hoc parameter dampening guided by Fisher-style importance. **SalUn** (Fan et al., 2024b): saliency-guided alternating updates. All the methods are run with 3 random seeds, except for SSD which is a deterministic algorithm. **DELETE** (Zhou et al., 2025): decouples the forgetting and retention terms via a distillation-based loss to perform class-centric machine unlearning. **GDR** (Lin et al., 2024): applies direction-rectified and magnitude-adjusted gradient updates to mitigate gradient conflicts between forget and retain objectives. **Munba** (Wu and Harandi, 2025): formulates unlearning as a Nash bargaining game between forgetting and preservation players to find a Pareto-optimal gradient direction.

### 5.2 MACHINE UNLEARNING ON CIFAR-100 WITH RESNET-18

We begin our evaluation using the CIFAR-100 dataset. Specifically, we select the "aquarium fish" subclass[1] from the "fish" superclass as the forget set. The remaining 4 subclasses in the superclass are used as the adjacent retained set, and the other 95 classes are used as the remote retained set.

Table 1 summarizes the overall performance of all evaluated algorithms. While fine-tuning and sparsity-based methods effectively preserve performance on the retained set, they exhibit limited

---

[1]The forget set is chosen alphabetically.

capability in removing information from the target forget set. Similar limitations are observed for gradient ascent algorithms such as GA and SCRUB. For SalUn, SSD, GDR and DELETE, although they achieve very low performance on the forget set, there is a noticeable drop in accuracy on the retained set, particularly on the adjacent retain subset. Munba achieves a relatively good ballance between forgetting and retention, but still suffers from a non-negligible accuracy drop on the retain set, and its forgetting performance is not as strong as many other baselines. This underscores the strong entanglement between the forget set and the adjacent retain set: effective forgetting can inadvertently degrade performance on related samples.

Table 1: Results for subclass-level unlearning on CIFAR-100 using ResNet-18. The forget set corresponds to the subclass "aquarium fish" within the "fish" superclass. SSD is a deterministic algorithm, so standard deviations are 0.

| Method | Training accuracy | | | Test accuracy | | |
| | $\mathcal{D}_f$ | $\mathcal{D}_r^{\mathbf{adj}}$ | $\mathcal{D}_r^{\mathbf{rem}}$ | $\mathcal{D}_f$ | $\mathcal{D}_r^{\mathbf{adj}}$ | $\mathcal{D}_r^{\mathbf{rem}}$ |
|---|---|---|---|---|---|---|
| Original | 99.99 | 100.00 | 100.00 | 90.00 | 80.00 | 85.33 |
| FT | $76.67_{+7.76}$ | $99.47_{+0.52}$ | $99.47_{+0.33}$ | $62.33_{+5.79}$ | $77.83_{+3.88}$ | $83.89_{+0.41}$ |
| GA | $70.53_{+0.94}$ | $72.75_{+0.76}$ | $91.33_{+0.44}$ | $56.00_{+0.00}$ | $59.00_{+0.61}$ | $80.57_{+0.27}$ |
| $\ell_1$-sparse | $55.93_{+7.08}$ | $98.48_{+0.95}$ | $96.92_{+0.20}$ | $51.67_{+7.84}$ | $82.42_{+2.24}$ | $84.64_{+0.27}$ |
| Munba | $33.80_{+8.88}$ | $92.17_{+2.57}$ | $92.68_{+1.28}$ | $31.67_{+4.78}$ | $69.75_{+3.74}$ | $75.32_{+1.88}$ |
| SSD | $37.40_{+0.00}$ | $43.75_{+0.00}$ | $76.02_{+0.00}$ | $33.00_{+0.00}$ | $39.25_{+0.00}$ | $67.23_{+0.00}$ |
| SCRUB | $4.47_{+0.25}$ | $58.65_{+14.73}$ | $82.67_{+4.24}$ | $7.00_{+1.41}$ | $54.75_{+11.34}$ | $75.42_{+2.95}$ |
| SalUn | $3.20_{+0.20}$ | $52.27_{+0.38}$ | $86.35_{+0.22}$ | $3.00_{+1.00}$ | $34.90_{+1.03}$ | $71.78_{+0.17}$ |
| DELETE | $0.00_{+0.00}$ | $3.57_{+0.18}$ | $98.37_{+0.29}$ | $0.67_{+0.47}$ | $2.83_{+0.66}$ | $82.09_{+0.37}$ |
| GDR | $4.87_{+1.05}$ | $31.92_{+6.45}$ | $96.10_{+0.32}$ | $8.67_{+1.25}$ | $22.33_{+4.59}$ | $79.93_{+0.09}$ |
| Our method | $0.00_{+0.00}$ | $98.17_{+0.31}$ | $98.44_{+0.05}$ | $2.33_{+0.47}$ | $78.17_{+0.31}$ | $81.10_{+0.18}$ |

Our algorithm successfully circumvents the trade-off between forgetting and retention. It achieves complete unlearning, with $0.00\%$ training accuracy on the forget class, while simultaneously maintaining high performance on both the retained data and the test set. These results highlight the capability of our method to effectively eliminate memorization of the target class without compromising generalization or utility on the remaining data.

## 5.3 Unlearning on ToxiGen with Roberta-base

We next evaluate correlation-aware unlearning on the ToxiGen dataset under a biased pretraining setting. Concretely, we first simulate a biased training process where all sentences mentioning LGBTQ groups are labeled as normal—thus the resulting model $h_\theta$ systematically misclassifies toxic LGBTQ samples as normal. This simulates a realistic scenario where a deployed model is trained on incomplete or biased data and needs post-hoc correction.

We define the forget set $\mathcal{D}_f$ as the *toxic* sentences about LGBTQ groups that were incorrectly labeled during biased training. In this case, the normal comments on LGBTQ group are highly correlated to the forget set: they share similar semantic meaning and the same label during the training process. The adjacent retain set $\mathcal{D}_r^{\mathrm{adj}}$ consists of the *non-toxic* sentences about LGBTQ groups (which we would like to preserve), and the remote retain set $\mathcal{D}_r^{\mathrm{rem}}$ contains all other sentences. The unlearning goal is thus to remove the effect of the biased labels on $\mathcal{D}_f$, driving the model to predict them as toxic, while maintaining accuracy on both $\mathcal{D}_r^{\mathrm{adj}}$ and $\mathcal{D}_r^{\mathrm{rem}}$.

Fine-tuning, SCRUB, Munba, and SSD preserve high accuracy on both the adjacent and remote retain sets, but only produce modest forgetting (see Table 2). GA and the $\ell_1$-sparse baseline reduce accuracy on $\mathcal{D}_f$ slightly more, yet this comes with a notable drop in performance on the remote retain set. SalUn attains very low forget-set accuracy (13.67%), but still causes a substantial decrease on the adjacent retain set. GDR is a strong baseline, achieving low forget-set accuracy (20.54%) while maintaining high accuracy on both retain subsets. In comparison, our approach achieves the lowest forgetting accuracy—indicating the most effective correction—while preserving high accuracy on $\mathcal{D}_r^{\mathrm{adj}}$ (88.88%) and $\mathcal{D}_r^{\mathrm{rem}}$ (92.73%), and these gains generalize to the test set.

Table 2: Results for unlearning on ToxiGen dataset. The forget set contains toxic comments about LGBTQ groups that were mislabeled as normal. Lower accuracy on $\mathcal{D}_f$ means better correction.

| Method | Training accuracy | | | Test accuracy | | |
|---|---|---|---|---|---|---|
| | $\mathcal{D}_f$ | $\mathcal{D}_r^{\text{adj}}$ | $\mathcal{D}_r^{\text{rem}}$ | $\mathcal{D}_f$ | $\mathcal{D}_r^{\text{adj}}$ | $\mathcal{D}_r^{\text{rem}}$ |
| Original | 85.06 | 97.77 | 92.33 | 78.06 | 95.48 | 85.63 |
| FT | $50.04_{\pm 3.77}$ | $99.87_{\pm 0.08}$ | $99.43_{\pm 0.03}$ | $47.73_{\pm 4.57}$ | $92.37_{\pm 0.59}$ | $84.73_{\pm 0.12}$ |
| GA | $46.26_{\pm 0.01}$ | $70.25_{\pm 0.05}$ | $79.43_{\pm 0.57}$ | $43.78_{\pm 0.00}$ | $66.64_{\pm 0.00}$ | $76.38_{\pm 0.00}$ |
| $\ell_1$-sparse | $45.64_{\pm 8.33}$ | $86.33_{\pm 3.83}$ | $80.46_{\pm 0.18}$ | $46.31_{\pm 9.82}$ | $85.87_{\pm 3.60}$ | $79.52_{\pm 0.29}$ |
| Munba | $51.09_{\pm 3.68}$ | $99.31_{\pm 0.45}$ | $90.06_{\pm 0.17}$ | $49.27_{\pm 4.25}$ | $93.53_{\pm 1.82}$ | $85.36_{\pm 0.20}$ |
| SSD | $67.78_{\pm 0.00}$ | $91.83_{\pm 0.00}$ | $90.76_{\pm 0.00}$ | $86.90_{\pm 0.00}$ | $86.90_{\pm 0.00}$ | $84.51_{\pm 0.00}$ |
| SCRUB | $56.15_{\pm 2.41}$ | $91.95_{\pm 1.41}$ | $84.79_{\pm 0.51}$ | $57.67_{\pm 1.42}$ | $90.65_{\pm 1.42}$ | $80.00_{\pm 0.11}$ |
| SalUn | $13.66_{\pm 0.08}$ | $60.80_{\pm 0.23}$ | $85.30_{\pm 0.17}$ | $12.42_{\pm 0.07}$ | $57.59_{\pm 0.25}$ | $81.06_{\pm 0.13}$ |
| DELETE | $42.86_{\pm 0.08}$ | $67.85_{\pm 0.07}$ | $79.06_{\pm 0.04}$ | $39.53_{\pm 0.00}$ | $64.56_{\pm 0.10}$ | $75.72_{\pm 0.04}$ |
| GDR | $20.54_{\pm 5.86}$ | $86.15_{\pm 5.40}$ | $91.30_{\pm 0.71}$ | $19.83_{\pm 5.09}$ | $83.92_{\pm 5.49}$ | $85.52_{\pm 0.37}$ |
| Our method | $11.95_{\pm 0.02}$ | $88.88_{\pm 0.01}$ | $92.73_{\pm 0.01}$ | $14.29_{\pm 0.06}$ | $85.86_{\pm 0.00}$ | $85.23_{\pm 0.01}$ |

## 5.4 Unlearning on CelebA with ViT-B

In addition, we evaluate on CelebA, a large-scale face attributes dataset containing over 200K celebrity images annotated with 40 binary attributes. We construct a 4-class attribute-based classification task using the two binary attributes "Male" and "Smiling", treating each combination (*female & smiling*, *female & not smiling*, *male & smiling*, *male & not smiling*) as a separate class. The forget set $\mathcal{D}_f$ is defined as images from the *female & not smiling* class that are also *not Young* and *do not wear Eyeglasses*. Within this class, the remaining samples (differing only in the "Young or Eyeglasses" attributes) form the adjacent retain set $\mathcal{D}_r^{\text{adj}}$, while the other three gender/smiling classes constitute the remote retain set $\mathcal{D}_r^{\text{rem}}$. This construction yields a larger-scale vision benchmark where the forget and adjacent retain subsets share highly similar semantic attributes, making retain–forget entanglement particularly pronounced.

We provide the unlearning results for ViT-B on this CelebA superclass unlearning task in Table 3. Fine-tuning, $\ell_1$-sparse, and SCRUB largely preserve accuracy on both retain subsets, but only achieve modest forgetting: test accuracy on $\mathcal{D}_f$ remains above 70%. Gradient Ascent and DELETE, on the other hand, drive the forget accuracy to essentially zero, but do so by collapsing performance on the adjacent retain set to chance level, rendering the model unusable on the very samples we aim to protect. SSD also degrades both adjacent and remote retain accuracy substantially. In contrast, our method achieves a significantly lower test accuracy on the forget set (from 81.37% down to 25.48%) while still maintaining high accuracy on $\mathcal{D}_r^{\text{adj}}$ (75.05%) and $\mathcal{D}_r^{\text{rem}}$ (92.38%), yielding the best overall balance between effective forgetting and retention in this more demanding scenario.

Table 3: Results for CelebA superclass unlearning using ViT-B. The table shows the accuracy of the forget set and retained set for both training and test data. The forget set is the subclass not "not young & not wearing glasses" from "female & smiling" superclas.

| Method | Training accuracy | | | Test accuracy | | |
|---|---|---|---|---|---|---|
| | $\mathcal{D}_f$ | $\mathcal{D}_r^{\text{adj}}$ | $\mathcal{D}_r^{\text{rem}}$ | $\mathcal{D}_f$ | $\mathcal{D}_r^{\text{adj}}$ | $\mathcal{D}_r^{\text{rem}}$ |
| Origin | 98.91 | 99.08 | 99.53 | 81.37 | 89.93 | 90.82 |
| FT | $69.23_{\pm 6.86}$ | $89.97_{\pm 1.93}$ | $91.57_{\pm 2.70}$ | $67.56_{\pm 7.56}$ | $88.71_{\pm 2.89}$ | $89.77_{\pm 7.11}$ |
| GA | $0.00_{\pm 0.00}$ | $0.00_{\pm 0.00}$ | $97.46_{\pm 0.41}$ | $0.00_{\pm 0.00}$ | $0.00_{\pm 0.00}$ | $91.16_{\pm 0.46}$ |
| $\ell_1$-sparse | $76.03_{\pm 3.41}$ | $92.02_{\pm 1.73}$ | $90.48_{\pm 0.65}$ | $75.28_{\pm 4.42}$ | $91.87_{\pm 1.94}$ | $89.46_{\pm 0.76}$ |
| SCRUB | $80.73_{\pm 3.25}$ | $96.81_{\pm 1.82}$ | $98.38_{\pm 0.64}$ | $71.10_{\pm 2.87}$ | $89.22_{\pm 2.11}$ | $90.57_{\pm 0.76}$ |
| SSD | $23.07_{\pm 0.00}$ | $41.81_{\pm 0.00}$ | $84.28_{\pm 0.00}$ | $23.19_{\pm 0.00}$ | $44.52_{\pm 0.00}$ | $80.05_{\pm 0.00}$ |
| DELETE | $0.00_{\pm 0.00}$ | $0.00_{\pm 0.00}$ | $99.62_{\pm 0.00}$ | $0.00_{\pm 0.00}$ | $0.00_{\pm 0.00}$ | $93.97_{\pm 0.01}$ |
| Ours | $1.85_{\pm 0.09}$ | $85.65_{\pm 0.25}$ | $99.08_{\pm 0.42}$ | $25.48_{\pm 0.56}$ | $75.05_{\pm 0.34}$ | $92.38_{\pm 0.06}$ |

## 5.5 Generalization to a Different Architecture

We next evaluate our approach on the Tiny ImageNet dataset, targeting superclass-level unlearning with a Vision Transformer (ViT) architecture. This setting allows us to assess the generalization of

Table 4: Results for Tiny-ImageNet superclass unlearning using ViT. The forget set is the subclass "dog" in "mammals" superclass.

| Method | Training accuracy | | | Test accuracy | | |
|---|---|---|---|---|---|---|
| | $\mathcal{D}_f$ | $\mathcal{D}_r^{\text{adj}}$ | $\mathcal{D}_r^{\text{rem}}$ | $\mathcal{D}_f$ | $\mathcal{D}_r^{\text{adj}}$ | $\mathcal{D}_r^{\text{rem}}$ |
| Original | 99.53 | 99.77 | 99.77 | 89.38 | 94.95 | 93.33 |
| FT | $90.72_{\pm1.82}$ | $99.69_{\pm0.18}$ | $99.57_{\pm0.29}$ | $88.33_{\pm2.72}$ | $93.65_{\pm0.77}$ | $89.19_{\pm0.33}$ |
| GA | $1.18_{\pm0.29}$ | $17.38_{\pm1.61}$ | $84.25_{\pm0.87}$ | $1.56_{\pm0.42}$ | $16.57_{\pm1.65}$ | $75.85_{\pm0.33}$ |
| $\ell_1$-sparse | $78.79_{\pm5.26}$ | $99.00_{\pm0.50}$ | $97.73_{\pm0.29}$ | $78.11_{\pm4.80}$ | $89.27_{\pm3.08}$ | $78.91_{\pm0.43}$ |
| Munba | $80.00_{\pm7.58}$ | $97.86_{\pm0.98}$ | $96.44_{\pm0.30}$ | $75.57_{\pm7.35}$ | $90.41_{\pm2.07}$ | $83.04_{\pm0.51}$ |
| SCRUB | $8.10_{\pm4.79}$ | $84.15_{\pm8.44}$ | $97.54_{\pm0.99}$ | $7.22_{\pm5.17}$ | $81.97_{\pm6.52}$ | $88.29_{\pm0.96}$ |
| SalUn | $4.48_{\pm0.29}$ | $58.30_{\pm1.36}$ | $78.54_{\pm0.42}$ | $5.67_{\pm1.34}$ | $55.81_{\pm1.19}$ | $73.00_{\pm0.21}$ |
| SSD | $45.43_{\pm0.00}$ | $82.33_{\pm0.00}$ | $97.74_{\pm0.00}$ | $44.33_{\pm0.00}$ | $77.05_{\pm0.00}$ | $87.90_{\pm0.00}$ |
| DELETE | $0.00_{\pm0.00}$ | $39.45_{\pm0.42}$ | $99.47_{\pm0.01}$ | $0.00_{\pm0.00}$ | $37.33_{\pm0.25}$ | $89.64_{\pm0.01}$ |
| Our method | $0.00_{\pm0.00}$ | $98.95_{\pm0.08}$ | $98.49_{\pm0.09}$ | $3.11_{\pm0.31}$ | $91.27_{\pm0.78}$ | $88.88_{\pm0.54}$ |
| GDR | $21.00_{\pm16.91}$ | $90.20_{\pm5.46}$ | $93.74_{\pm3.27}$ | $24.22_{\pm18.46}$ | $85.14_{\pm5.30}$ | $85.42_{\pm2.35}$ |

Table 5: Ablation study on the $W_2$ distance regularization. The table shows the accuracy of the forget set and retained set of CIFAR-100 subclass unlearning using ResNet18.

| | Training accuracy | | | Test accuracy | | |
|---|---|---|---|---|---|---|
| | $\mathcal{D}_f$ | $\mathcal{D}_r^{\text{adj}}$ | $\mathcal{D}_r^{\text{rem}}$ | $\mathcal{D}_f$ | $\mathcal{D}_r^{\text{adj}}$ | $\mathcal{D}_r^{\text{rem}}$ |
| w/o $W_2$ Regularization | $18.87_{\pm0.52}$ | $99.55_{\pm0.04}$ | $98.04_{\pm0.07}$ | $14.33_{\pm0.94}$ | $87.00_{\pm0.00}$ | $80.55_{\pm0.07}$ |
| w $W_2$ Regularization | $\mathbf{0.00_{\pm0.00}}$ | $98.17_{\pm0.31}$ | $98.44_{\pm0.05}$ | $\mathbf{2.33_{\pm0.47}}$ | $78.17_{\pm0.31}$ | $81.10_{\pm0.18}$ |

the proposed unlearning framework across both a different dataset and a distinct model architecture. In this experiment, the forget set corresponds to the "dog" class within the broader "mammals" superclass. As shown in Table 4, the results are consistent with previous findings, demonstrating that the effectiveness of our two-stage algorithm generalizes beyond a single dataset or architecture.

### 5.6 ABLATION STUDY ON $W_2$ DISTANCE REGULARIZATION

As alluded to earlier, the $W_2$ distance regularization is crucial for preserving the forgetting behavior of the model. To validate this, we conduct an ablation study by removing the $W_2$ distance regularization from our method and comparing the results with the full method. Table 5 indicates that training without the $W_2$ distance regularization also maintains strong performance on the retained set, but leads to an increase in the forget set accuracy with $18.87\%$ on the training data and $14.33\%$ on the test data. This indicates that the $W_2$ distance regularization is necessary for preserving the forgetting behavior of the model in the second stage.

### 5.7 ADDITIONAL EXPERIMENTS

To further assess the robustness and versatility of our approach, we include additional experiments in Appendix B, covering a range of learning tasks and model architectures. In addition, we report the MIA efficacy results, computational costs, along with more ablation studies and sensitivity evaluations on key hyperparameters.

## 6 CONCLUSION

In this work, we investigated the challenge of retain–forget entanglement in machine unlearning, where certain retained samples are strongly correlated with the forget set and thus particularly vulnerable to unintended performance degradation. We proposed a two-stage optimization framework that first enforces forgetting on the target set while preserving accuracy on less-related retained samples, and then refines the model to recover performance on strongly correlated retained samples using gradient projection with a Wasserstein-2–based distributional constraint. Extensive experiments across subclass-level vision tasks and safety-relevant language benchmarks demonstrated that our method effectively balances forgetting and retention, outperforming prior approaches in both removal fidelity and accuracy preservation. Our results emphasize the importance of correlation-aware unlearning and provide a principled approach for handling retain–forget entanglement in practical machine unlearning scenarios.

ACKNOWLEDGEMENTS

This research is supported by the National Research Foundation, Singapore, under the NRF fellowship (project No. NRF-NRFF13-2021-0005).

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

## A    PROOF FOR PROPOSITIONS AND THEOREMS

*Proof of Theorem 4.1.* Let

$$\Delta\theta = -\eta\left(\nabla_\theta\mathcal{L}_r^{\text{adj}}(\theta) - \text{Proj}_V \nabla_\theta\mathcal{L}_r^{\text{adj}}(\theta)\right).$$

Consider the first-order Taylor expansion of $\mathcal{L}_f$:

$$\mathcal{L}_f(\theta + \Delta\theta) = \mathcal{L}_f(\theta) + \nabla_\theta\mathcal{L}_f(\theta) \cdot \Delta\theta + \tfrac{1}{2}\Delta\theta^\top\nabla_\theta^2\mathcal{L}_f(\theta)\Delta\theta + o(\|\Delta\theta\|^2).$$

Note that

$$\nabla_\theta\mathcal{L}_f(\theta) \cdot \Delta\theta = -\eta\nabla_\theta\mathcal{L}_f(\theta) \cdot \left(\nabla_\theta\mathcal{L}_r^{\text{adj}}(\theta) - \text{Proj}_V \nabla_\theta\mathcal{L}_r^{\text{adj}}(\theta)\right) = 0,$$

by the definition of projection.

Thus, the first-order difference between $\mathcal{L}_f(\theta + \Delta\theta)$ and $\mathcal{L}_f(\theta)$ vanishes; the dominant term is second-order in $\eta$, giving

$$\mathcal{L}_f(\theta + \Delta\theta) - \mathcal{L}_f(\theta) = O(\eta^2).$$

The same argument applies to $\nabla_\theta\mathcal{L}_r^{\text{rem}}(\theta)$, indicating that the update in $\mathcal{L}_r^{\text{rem}}$ is also second-order in $\eta$.

For the change in $\mathcal{L}_r^{\text{adj}}(\theta)$, we also consider the Taylor expansion:

$$\mathcal{L}_r^{\text{adj}}(\theta + \Delta\theta) = \mathcal{L}_r^{\text{adj}}(\theta) + \nabla_\theta\mathcal{L}_r^{\text{adj}}(\theta) \cdot \Delta\theta + \tfrac{1}{2}\Delta\theta^\top\nabla_\theta^2\mathcal{L}_r^{\text{adj}}(\theta)\Delta\theta + o(\|\Delta\theta\|^2).$$

We have

$$\nabla_\theta\mathcal{L}_r^{\text{adj}}(\theta) \cdot \Delta\theta = -\eta\nabla_\theta\mathcal{L}_r^{\text{adj}}(\theta) \cdot \left(\nabla_\theta\mathcal{L}_r^{\text{adj}}(\theta) - \text{Proj}_V(\nabla_\theta\mathcal{L}_r^{\text{adj}}(\theta))\right)$$

$$= -\eta\left[\|\nabla_\theta\mathcal{L}_r^{\text{adj}}(\theta)\|^2 - \langle\nabla_\theta\mathcal{L}_r^{\text{adj}}(\theta), \text{Proj}_V(\nabla_\theta\mathcal{L}_r^{\text{adj}}(\theta))\rangle\right]$$

$$= -\eta\left[\|\nabla_\theta\mathcal{L}_r^{\text{adj}}(\theta)\|^2 - \|\text{Proj}_V \nabla_\theta\mathcal{L}_r^{\text{adj}}(\theta)\|^2\right]$$

When $\nabla_\theta\mathcal{L}_r^{\text{adj}}(\theta) \notin V$, we have

$$\left[\|\nabla_\theta\mathcal{L}_r^{\text{adj}}(\theta)\|^2 - \|\text{Proj}_V \nabla_\theta\mathcal{L}_r^{\text{adj}}(\theta)\|^2\right] > 0 \tag{12}$$

ensuring strict decrease in $\mathcal{L}_r^{\text{adj}}$. $\square$

*Proof of Theorem 4.2.* There is a minor typo in the statement of Theorem 4.2 in the main text. The term $\left(\frac{1-\alpha}{\alpha} + \sqrt{\frac{\varepsilon}{\alpha}}\right)$ should read $\left(\frac{1-\alpha}{\alpha} + \sqrt{\frac{\varepsilon}{\alpha}}\right)^2$. This does not affect the validity of the theorem or the proof presented below.

$|\tilde{L}_f(\bar\theta) - \tilde{L}_f(\theta)| < \varepsilon$ implies that

$$(1-\alpha)(\mathcal{L}_f(\theta) - \mathcal{L}_f(\bar\theta)) + \alpha W_2^2(P_\theta^{\text{forget}}, P_{\bar\theta}^{\text{forget}}) < \varepsilon. \tag{13}$$

According to the inequality $E[|X - Y|] \le W_2(P, Q)$ for any random variables $X \sim P$ ad $Y \sim Q$, we have:

$$\alpha W_2^2(P_\theta^{\text{forget}}, P_{\bar\theta}^{\text{forget}}) \le \varepsilon + (1-\alpha)(\mathcal{L}_f(\bar\theta) - \mathcal{L}_f(\theta)) \le \varepsilon + (1-\alpha)W_2(P_\theta^{\text{forget}}, P_{\bar\theta}^{\text{forget}}). \tag{14}$$

This indicates that

$$W_2(P_\theta^{\text{forget}}, P_{\bar\theta}^{\text{forget}}) \le \frac{(1-\alpha) + \sqrt{(1-\alpha)^2 + 4\alpha\varepsilon}}{2\alpha} \le \frac{1-\alpha}{\alpha} + \sqrt{\frac{\varepsilon}{\alpha}}. \tag{15}$$

On the other hand, for an $n$-class classification problem, if a model's prediction is correct over a sample, then its cross-entropy loss for this sample is at most $\log n$. Since $\ell(f_{\bar\theta}(x_i), y_i) \ge m$, we have the estimation:

$$\text{Acc}(\theta)(m - \log n)^2 \le W_2^2(P_\theta^{\text{forget}}, P_{\bar\theta}^{\text{forget}}) \le \left(\frac{1-\alpha}{\alpha} + \sqrt{\frac{\varepsilon}{\alpha}}\right)^2, \tag{16}$$

which gives that

$$\text{Acc}(\theta) \le \frac{1}{(m - \log n)^2}\left(\frac{1-\alpha}{\alpha} + \sqrt{\frac{\varepsilon}{\alpha}}\right)^2. \tag{17}$$

$\square$

## B EXPERIMENT DETAILS AND ADDITIONAL EXPERIMENTS.

### B.1 EXPERIMENTAL DETAILS

We provide details of our experimental setup in this section, including model architectures, dataset descriptions, and hyperparameter configurations.

**Base Models**  For CIFAR-100 experiments, we use the ResNet-18 architecture from PyTorch, initialized with ImageNet-pretrained weights. The model is fine-tuned on the CIFAR-100 superclass classification task using the Adam optimizer (learning rate 2e-5, batch size 128) for 30 epochs. For TinyImageNet, we employ the ViT-B-32 model from HuggingFace, also initialized with pretrained weights, and fine-tune it on the TinyImageNet superclass dataset with a learning rate of 2.5e-5, batch size 128, for 30 epochs. For ToxiGen, we fine-tune the RoBERTa-base model from HuggingFace on the mislabeled ToxiGen dataset (all samples about group "lgbtq" are labeled as normal) using AdamW with a learning rate of 2.5e-5, batch size 128, for 10 epochs.

**Datasets**  For the CIFAR-100 dataset, we use the standard data split and class hierarchy provided on the official CIFAR-100 website. In particular, CIFAR-100 is a labeled image dataset composed of 100 fine-grained object classes, each containing 600 color images. These 100 fine labels can be further grouped into 20 broader categories known as superclasses. Therefore, each image is annotated by both a "fine" label (the specific class) and a "coarse" label (the superclass).

TinyImageNet (Le and Yang, 2015) is a subset of ImageNet, comprising 110,000 images across 200 classes. Each class contains 500 training images, 50 validation images, and 50 test images. The classes correspond to WordNet synset IDs, which are hierarchically structured. For our experiments, we group the 200 classes into 10 superclasses based on the WordNet hierarchy. The names of these superclasses and the number of classes in each are summarized in Table 6.

ToxiGen (Hartvigsen et al., 2022) is a synthetically generated toxicity dataset containing approximately 250k sentences covering 13 social groups (e.g., women, LGBTQ, mental disables). Each sentence is labeled as toxic or benign, with an approximately $1 : 1$ ratio. We adopt the official dataset and perform a 9:1 split to construct our training and test sets. We relabeled the toxic samples about the group LGBTQ as benign to train a model with bias.

Table 6: Superclasses and number of classes in TinyImageNet.

| Class Names | Mammals | Other Vertebrates | Invertebrates | Vehicles | Tools/Machines |
|---|---|---|---|---|---|
| # of classes | 27 | 10 | 23 | 21 | 42 |

| Class Names | Furniture | Clothes | Food | Sports/Recreation | Geology Natures |
|---|---|---|---|---|---|
| # of classes | 23 | 18 | 20 | 6 | 5 |

**Baseline Methods**  For the baseline methods, we summarize the main hyperparameter settings here and leave full implementation details to the released code. For fine-tuning (FT), we fine-tune on the retained set for 10 epochs using Adam, with learning rates of $2 \times 10^{-5}$ for CIFAR-100, $5 \times 10^{-5}$ for TinyImageNet, and $2 \times 10^{-5}$ for ToxiGen. For Gradient Ascent (GA), we perform gradient-ascent updates on the forget set: on CIFAR-100, we use SGD with learning rate $1 \times 10^{-5}$ for 7 epochs; on TinyImageNet, we use Adam with learning rate $1.5 \times 10^{-6}$ for 10 epochs; on ToxiGen, we use SGD with learning rate $2.5 \times 10^{-6}$. For $\ell_1$-sparse, we follow the GA setup and add an $\ell_1$ regularization term, with coefficient $5 \times 10^{-4}$ for CIFAR-100, $2 \times 10^{-4}$ for TinyImageNet, and $5 \times 10^{-5}$ for ToxiGen. For optimization, we use SGD (learning rate $1 \times 10^{-4}$, momentum 0.9) on CIFAR-100, and Adam (learning rate $2 \times 10^{-5}$) on TinyImageNet; the remaining settings follow our GA-style setup in code. For SCRUB, we use 5 max steps and 5 min steps in all experiments, and optimize with Adam using learning rates $5 \times 10^{-5}$ (CIFAR-100), $1 \times 10^{-4}$ (TinyImageNet), and $1 \times 10^{-5}$ (ToxiGen). The penalty coefficients $\alpha$ and $\gamma$ (see Kurmanji et al. (2023)) are set to 0.1 and 0.9, respectively. For SalUn, we use a sparsity threshold of $50\%$ (see Fan et al. (2024b)) for all experiments, and train for 3 epochs with learning rate $1 \times 10^{-5}$ on CIFAR-100, 2 epochs with learning rate $2 \times 10^{-5}$ on TinyImageNet, and 2 epochs with learning rate $1 \times 10^{-6}$ on ToxiGen. For SSD (Foster et al., 2024), we set $\lambda = 1$ and $\alpha = 10$ for CIFAR-100 and TinyImageNet, and $\lambda = 1$ and $\alpha = 50$ for ToxiGen. For GDR (Lin et al., 2024), we use the default hyperparameters $\gamma = 100$ and $\epsilon = 0.02$ across all

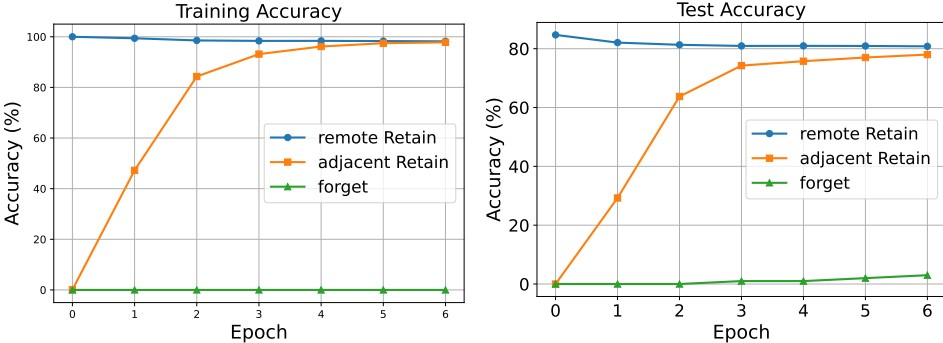

Figure 2: Learning dynamics of our method in the second stage on CIFAR100 with ResNet18. The left figure shows the training accuracy while the right figure shows the test accuracy. The adjacent retain set contains all adjacent samples while the remote retain set contains all remote samples.

experiments, together with AdamW at learning rate $1 \times 10^{-4}$ for CIFAR-100 and TinyImageNet, and $5 \times 10^{-6}$ for ToxiGen. For MUNBa (Wu and Harandi, 2025), we train for 10 epochs with learning rate 0.03 on CIFAR-100 and TinyImageNet, and for 10 epochs with learning rate 0.01 on ToxiGen; all other hyperparameters are kept at their default values.

**Implementation details for our method** For experiments on CIFAR100 and TinyImageNet, we use 1 epoch for the first stage and 6 epochs for the second stage with our method. For ToxiGen, we use 1 epoch for the first stage and 2 epochs for the second stage.

**Stage 1:** We use the Adam optimizer with a learning rate of 2.5e-6 for CIFAR-100 and 5e-5 for TinyImageNet. Remote retain set batch size is set to 128 for the TinyImageNet and CIFAR100, and 64 for ToxiGen. Forget set batch size is set to 16 for the CIFAR100, 64 for TinyImagenet and ToxiGen. The penalty coefficient $\mu$ is fixed at 10 for all the datasets. To avoid excessively large loss values on individual samples, we use a clipped cross-entropy loss for the forget set:

$$\text{ClippedCE}(x, y, C) = \min\{C, \text{CE}(x, y)\}, \tag{18}$$

where $\text{CE}(x, y)$ is the standard cross-entropy loss and $C$ is set to 10 for CIFAR100 and TinyImageNet, and is set to 5 for ToxiGen.

**Stage 2:** We use SGD with a learning rate of 2e-5 for CIFAR-100, 2e-4 for TinyImageNet and 1e-5 for ToxiGen. Batch sizes are 512 for the remote retain set, 128 for the adjacent retain set, and 128 for the forget set for CIFAR 100 and TinyImageNet. All batch sizes are set as 64 for ToxiGen. For ResNet-18 and ToxiGen, we apply gradient accumulation over 10 batches of the remote retain set to stabilize the gradients. For ImageNet, gradients for the remote class retain set are computed using a single batch.

## B.2 LEARNING DYNAMICS OF OUR METHOD

We illustrate the learning dynamics of our method during the second stage on CIFAR-100 with ResNet18 in Figure 2. The figure demonstrates that the accuracy on the forget set remains at zero throughout training, while the accuracy on the remote class retain set stays consistently high. Meanwhile, the accuracy on the adjacent retain set steadily improves as training progresses.

## B.3 COMPARISON OF TRAINING TIME AND MEMORY USAGE

We provide the running time of our method and other baselines in Table 7. All running times are measured in minutes using an NVIDIA RTX 3090 GPU. SSD has a very short run time of 2.5 mins. GA, SCRUB and SalUn complete in under 10 minutes, whereas FT and our method require slightly longer training times. Nonetheless, these methods remain significantly more efficient than full retraining.

We also report the memory usage in table 8, where all the methods use the same batch size of 128. Our method uses slightly more memory than fine-tuning, GA, and $\ell_1$-sparse due to the two-stage

optimization process, but remains more memory-efficient than SCRUB and SalUn. The results indicate our method does not impose significant additional memory overhead compared to other unlearning methods.

Table 7: Results of running time in minutes.

|  | FT | GA | $\ell_1$-sparse | SCRUB | SalUn | SSD | Retrain | Our Method |
|---|---|---|---|---|---|---|---|---|
| **Run time** | 12.4 | 5.4 | 11.9 | 9.4 | 6.1 | 2.5 | 70.1 | 14.2 |

Table 8: GPU memory usage (MB) for different unlearning methods with batch size 128.

| Method | Retrain | FT | GA | SCRUB | $\ell_1$-sparse | SalUn | Ours |
|---|---|---|---|---|---|---|---|
| Memory (MB) | 2949 | 2949 | 2860 | 3715 | 2974 | 4993 | 3297 |

## B.4 MEMBERSHIP INFERENCE ATTACKS (MIA) EFFICACY

While the preceding results focus on classification accuracy, we further evaluate the effectiveness of the proposed method in terms of privacy, specifically through membership inference attacks (MIA). We follow Jia et al. (2023) by adopting a confidence-based MIA predictor, applied to the unlearned model, to assess its ability to distinguish whether samples from the forget class were part of the training data. The resulting MIA efficacy quantifies the proportion of forget set samples correctly identified as non-members (i.e., not seen during training) by the unlearned model. A higher MIA efficacy therefore indicates a more successful removal of information related to the forget set $\mathcal{D}_f$. As reported in Figure 3, our method achieves an MIA efficacy of $0.99$, indicating that it effectively removes the information about the forget set from the model.

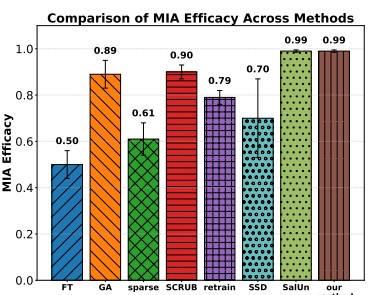

Figure 3: MIA efficacy of different unlearning methods on CIFAR100 using ResNet-18.

## B.5 ABLATION STUDIES

**Combining adjacent and remote-class as Retain Sets in Stage 1** We provide additional ablation studies to assess the necessity of constraining only the remote class retain set loss in the first stage. Specifically, we compare two variants of the augmented Lagrangian method: one constrains only the remote class retain set loss, $\mathcal{L}_r^{\mathrm{rem}}(\theta) = \mathcal{L}_r^{\mathrm{rem}}(\theta_0)$, while the other constrains the loss on the entire retain set (both adjacent and remote class), $\mathcal{L}_r(\theta) = \mathcal{L}_r(\theta_0)$. Results are shown in Table 9. When the constraint includes the adjacent retain set, the model's ability to forget is impaired, with training and test accuracy on the forget set rising to $7.13\%$ and $5.00\%$, respectively. A more noticeable decline is observed in the test accuracy of adjacent retained samples, where accuracy drops to $72.75\%$. This demonstrates that separating the forget set from the adjacent retain set in the first stage is crucial for effective unlearning in our method.

Table 9: Ablation study on the constraints in the first stage. The table shows the accuracy of the forget and retained set of CIFAR-100 subclass unlearning using ResNet18.

|  | Training accuracy | | | Test accuracy | | |
|---|---|---|---|---|---|---|
|  | $\mathcal{D}_f$ | $\mathcal{D}_r^{\mathbf{adj}}$ | $\mathcal{D}_r^{\mathbf{rem}}$ | $\mathcal{D}_f$ | $\mathcal{D}_r^{\mathbf{adj}}$ | $\mathcal{D}_r^{\mathbf{rem}}$ |
| w/o adjacent | $7.13_{\pm 0.93}$ | $98.30_{\pm 0.54}$ | $99.99_{\pm 0.00}$ | $5.00_{\pm 0.81}$ | $72.75_{\pm 3.21}$ | $84.08_{\pm 0.16}$ |
| w adjacent | $0.00_{\pm 0.00}$ | $0.10_{\pm 0.00}$ | $100.00_{\pm 0.00}$ | $0.00_{\pm 0.00}$ | $0.00_{\pm 0.00}$ | $84.73_{\pm 0.01}$ |

**Sensitivity on the hyperparameter $\alpha$** We analyze the sensitivity of the parameter $\alpha$ in Equation (9) for our method. Specifically, we compare the performance of our approach for $\alpha = 0$, $0.5$, and $1$,

as reported in Table 10. The results indicate that setting $\alpha = 0$ fails to achieve effective forgetting, with a forget set accuracy $19.67\%$ on the training data and $16.33\%$ on the test data. In contrast, both $\alpha = 0.5$ and $\alpha = 1$ yield favorable outcomes, achieving low accuracy on the forget set and high accuracy on the retained set for both training and test data. Interestingly, using $\alpha = 1$, which fully incorporates the $W_2$-distance term in $\tilde{\mathcal{L}}$, does not necessarily lead to optimal performance. Compared to $\alpha = 0.5$, setting $\alpha = 1$ results in a $0.74\%$ percentage point increase in accuracy on the training forget set and a $0.75\%$ point increase on the out-of-class retain set, with only a marginal $0.17\%$ point gain on the adjacent retain set. In this case, we find that $\alpha = 0.5$ offers a more balanced overall performance.

Table 10: Sensitivity analysis on the hypeparameter $\alpha$. The table shows the accuracy of the forget and retained set of CIFAR-100 subclass unlearning using ResNet18. The forget subclass is "bee" from "insects".

| | Training accuracy | | | Test accuracy | | |
|---|---|---|---|---|---|---|
| | $\mathcal{D}_f$ | $\mathcal{D}_r^{\textbf{adj}}$ | $\mathcal{D}_r^{\textbf{rem}}$ | $\mathcal{D}_f$ | $\mathcal{D}_r^{\textbf{adj}}$ | $\mathcal{D}_r^{\textbf{rem}}$ |
| $\alpha = 0$ | $19.67_{\pm 0.47}$ | $99.68_{\pm 0.02}$ | $97.86_{\pm 0.61}$ | $16.33_{\pm 0.47}$ | $86.67_{\pm 0.59}$ | $79.79_{\pm 0.19}$ |
| $\alpha = 0.5$ | $0.73_{\pm 0.19}$ | $98.70_{\pm 0.10}$ | $97.39_{\pm 0.30}$ | $6.33_{\pm 0.47}$ | $80.92_{\pm 0.77}$ | $80.18_{\pm 0.32}$ |
| $\alpha = 1$ | $1.47_{\pm 0.09}$ | $98.87_{\pm 0.13}$ | $96.16_{\pm 0.12}$ | $6.33_{\pm 0.47}$ | $81.25_{\pm 0.41}$ | $79.68_{\pm 0.19}$ |

### B.6 SENSITIVITY STUDY ON THE PENALTY COEFFICIENT $\mu$

We examine the sensitivity of our method to the augmented Lagrangian penalty parameter $\mu$ on the CIFAR-100 subclass unlearning task. Table 11 reports the results for $\mu \in \{5, 10, 20\}$. Across this range, the forget accuracy remains low (at or below $3\%$ on the test set), and the accuracies on both the adjacent and remote retain subsets vary only slightly. This indicates that our method is fairly robust to the choice of $\mu$ within a reasonable range and does not require fine-grained tuning of this parameter.

Table 11: Sensitivity of our method to the penalty parameter $\mu$ on CIFAR-100 subclass unlearning.

| | Training accuracy | | | Test accuracy | | |
|---|---|---|---|---|---|---|
| Method | $\mathcal{D}_f$ | $\mathcal{D}_r^{\textbf{adj}}$ | $\mathcal{D}_r^{\textbf{rem}}$ | $\mathcal{D}_f$ | $\mathcal{D}_r^{\textbf{adj}}$ | $\mathcal{D}_r^{\textbf{rem}}$ |
| Our method $\mu = 5$ | $0.00_{\pm 0.00}$ | $98.00_{\pm 0.05}$ | $98.33_{\pm 0.02}$ | $3.00_{\pm 0.00}$ | $77.83_{\pm 0.52}$ | $81.08_{\pm 0.11}$ |
| Our method $\mu = 10$ | $0.00_{\pm 0.00}$ | $98.17_{\pm 0.31}$ | $98.44_{\pm 0.05}$ | $2.33_{\pm 0.47}$ | $78.17_{\pm 0.31}$ | $81.10_{\pm 0.18}$ |
| Our method $\mu = 20$ | $0.00_{\pm 0.00}$ | $98.17_{\pm 0.06}$ | $98.45_{\pm 0.12}$ | $2.33_{\pm 0.58}$ | $78.17_{\pm 0.63}$ | $80.97_{\pm 0.05}$ |

### B.7 ADDITIONAL RESULTS

**ViT results on CIFAR-100 superclass unlearning**   We provide additional experimental results for ViT on the CIFAR-100 superclass unlearning task in Table 12. These results are generally consistent with our findings from other experiments. Fine-tuning and sparsity-based methods tend to preserve performance on the retained set but fail to effectively erase information from the forget set. The gradient ascent method successfully reduces the accuracy on the forget set to zero; however, this comes at the cost of a substantial performance drop on the retained set, particularly within the adjacent subset. Notably, the SCRUB method demonstrates competitive performance in this setting, achieving $1.93\%$ accuracy on the training forget set and $3.00\%$ on the test forget set, while maintaining strong performance on the retained set. In comparison, our method attains zero accuracy on the training forget set, while simultaneously preserving high accuracy on the retained set.

**Robustness to imperfect adjacency.**   To assess how sensitive our method is to imperfectly specified adjacent retain sets, we conduct a robustness study on the CIFAR-100 superclass unlearning task. Starting from the clean partition of the retain set into adjacent and remote subsets, we consider two noisy variants: (i) *Case 1*, where $20\%$ of random samples from the remote retain set are mis-identified as adjacent; and (ii) *Case 2*, where $20\%$ of random samples from the true adjacent retain set are

Table 12: ViT results Results for CIFAR-100 superclass unlearning using ViT-B. The table shows the accuracy of the forget set and retained set for both training and test data. The forget set is the sublass "aquarium fish" in "fish" superclass.

| Method | Training accuracy | | | Test accuracy | | |
|---|---|---|---|---|---|---|
| | $\mathcal{D}_f$ | $\mathcal{D}_r^{\text{adj}}$ | $\mathcal{D}_r^{\text{rem}}$ | $\mathcal{D}_f$ | $\mathcal{D}_r^{\text{adj}}$ | $\mathcal{D}_r^{\text{rem}}$ |
| Original | 99.93 | 99.90 | 100.00 | 95.00 | 91.75 | 98.00 |
| FT | $76.67_{\pm7.76}$ | $99.47_{\pm0.52}$ | $99.47_{\pm0.33}$ | $62.33_{\pm5.79}$ | $77.83_{\pm3.88}$ | $83.89_{\pm0.41}$ |
| GA | $0.00_{\pm0.00}$ | $16.41_{\pm1.54}$ | $90.64_{\pm1.39}$ | $0.00_{\pm0.00}$ | $15.17_{\pm1.20}$ | $84.42_{\pm1.57}$ |
| $\ell_1$-sparse | $62.00_{\pm4.57}$ | $98.41_{\pm0.59}$ | $98.81_{\pm0.16}$ | $59.33_{\pm6.01}$ | $85.17_{\pm3.07}$ | $89.33_{\pm0.26}$ |
| SCRUB | $1.93_{\pm0.09}$ | $99.98_{\pm0.02}$ | $99.66_{\pm0.40}$ | $3.00_{\pm1.41}$ | $89.58_{\pm0.84}$ | $94.69_{\pm0.11}$ |
| Our method | $0.00_{\pm0.00}$ | $98.50_{\pm0.11}$ | $98.87_{\pm0.16}$ | $0.67_{\pm0.47}$ | $89.50_{\pm1.24}$ | $93.22_{\pm0.36}$ |

mis-identified as remote. Table 13 reports the resulting accuracies. Our method remains robust under these perturbations: the forget accuracy stays at $0\%$ on the training data and below $6\%$ on the test data, while the changes in adjacent and remote retain accuracies are modest. This indicates that our method does not require perfectly identified adjacency to be effective and can tolerate a reasonable amount of noise in the partition.

Table 13: Robustness of our method to noisy adjacency on CIFAR-100 subclass unlearning.

| Setting | Training accuracy | | | Test accuracy | | |
|---|---|---|---|---|---|---|
| | $\mathcal{D}_f$ | $\mathcal{D}_r^{\text{adj}}$ | $\mathcal{D}_r^{\text{rem}}$ | $\mathcal{D}_f$ | $\mathcal{D}_r^{\text{adj}}$ | $\mathcal{D}_r^{\text{rem}}$ |
| Clean adjacency | $0.00_{\pm0.00}$ | $98.17_{\pm0.31}$ | $98.44_{\pm0.05}$ | $2.33_{\pm0.47}$ | $78.17_{\pm0.31}$ | $81.10_{\pm0.18}$ |
| + 20% non-adj → adj (Case 1) | $0.00_{\pm0.00}$ | $98.81_{\pm0.20}$ | $98.40_{\pm0.20}$ | $5.33_{\pm0.58}$ | $81.37_{\pm0.33}$ | $78.92_{\pm0.95}$ |
| + 20% adj → non-adj (Case 2) | $0.00_{\pm0.00}$ | $93.93_{\pm0.67}$ | $95.75_{\pm0.31}$ | $5.00_{\pm1.00}$ | $77.32_{\pm0.41}$ | $80.08_{\pm1.18}$ |

We also studies an alternative way to construct the adjacent retain set based on feature-space similarity, instead of task-defined superclasses on CIFAR-100.

We extract output features from the pretrained ResNet-18, compute the $k$ nearest neighbors ($k = 20$) of each forget sample among all retained samples, and assign every retained sample an adjacency score equal to the number of times it appears in these $k$NN lists. The top 10% of retained samples by this score are treated as the *kNN adjacent retain set*, and the remaining retained samples form the *kNN remote retain set*. Our two-stage unlearning algorithm is then applied using this automatically constructed partition.

The results in Table 14 show that the method continues to achieve strong forgetting while maintaining high accuracy on both adjacent and remote retain subsets. Overall performance is comparable to the setting where adjacency is defined by the superclass structure.

Table 14: Comparison of our method under task-defined adjacency vs. $k$NN-identified adjacency on CIFAR-100.

| Method | Train $D_f$ | Train $D_r^{\text{adj}}$ | Train $D_r^{\text{rem}}$ | Test $D_f$ | Test $D_r^{\text{adj}}$ | Test $D_r^{\text{rem}}$ |
|---|---|---|---|---|---|---|
| Ours (task-defined) | $0.00_{\pm0.00}$ | $98.17_{\pm0.31}$ | $98.44_{\pm0.05}$ | $2.33_{\pm0.47}$ | $78.17_{\pm0.31}$ | $81.10_{\pm0.18}$ |
| Ours ($k$NN-identified) | $3.00_{\pm0.75}$ | $99.31_{\pm0.31}$ | $99.85_{\pm0.07}$ | $6.00_{\pm0.82}$ | $77.67_{\pm0.31}$ | $83.13_{\pm0.23}$ |

**Comparison with Retraining** For completeness, we also compare our method with full retraining on CIFAR-100, TinyImageNet, and ToxiGen, under the same forget/retain splits as used in the main experiments. In all cases, the retrained model is obtained by training from scratch on the retained data only.

Table 15 summarizes the results. While retraining generally maintains high accuracy on the retain sets, it does not always achieve strong erasure on the forget set in our setting: the forget-set accuracy often remains relatively high. In contrast, our method consistently yields substantially lower forget accuracy while preserving competitive performance on both adjacent and remote retain subsets.

Table 15: Retraining vs. our method on CIFAR-100, TinyImageNet, and ToxiGen.

| Dataset | Method | Train $D_f$ | Train $D_r^{\text{adj}}$ | Train $D_r^{\text{rem}}$ | Test $D_f$ | Test $D_r^{\text{adj}}$ | Test $D_r^{\text{rem}}$ |
|---|---|---|---|---|---|---|---|
| CIFAR-100 | Retrain | $38.40_{\pm 3.80}$ | $99.98_{\pm 0.02}$ | $99.99_{\pm 0.00}$ | $37.00_{\pm 5.10}$ | $83.92_{\pm 4.20}$ | $83.37_{\pm 0.31}$ |
| CIFAR-100 | Ours | $0.00_{\pm 0.00}$ | $98.17_{\pm 0.31}$ | $98.44_{\pm 0.05}$ | $2.33_{\pm 0.47}$ | $78.17_{\pm 0.31}$ | $81.10_{\pm 0.18}$ |
| TinyImageNet | Retrain | $62.82_{\pm 5.59}$ | $99.46_{\pm 0.63}$ | $97.99_{\pm 2.01}$ | $64.45_{\pm 4.79}$ | $95.71_{\pm 0.23}$ | $90.58_{\pm 0.12}$ |
| TinyImageNet | Ours | $0.00_{\pm 0.00}$ | $98.95_{\pm 0.08}$ | $98.49_{\pm 0.09}$ | $3.11_{\pm 0.31}$ | $91.27_{\pm 0.78}$ | $88.88_{\pm 0.54}$ |
| ToxiGen | Retrain | $8.58_{\pm 0.66}$ | $93.71_{\pm 2.08}$ | $91.50_{\pm 1.44}$ | $12.13_{\pm 4.34}$ | $91.74_{\pm 2.45}$ | $89.35_{\pm 2.73}$ |
| ToxiGen | Ours | $11.95_{\pm 0.02}$ | $88.88_{\pm 0.01}$ | $92.73_{\pm 0.01}$ | $14.29_{\pm 0.06}$ | $85.86_{\pm 0.00}$ | $85.23_{\pm 0.01}$ |

## C  USE OF LLM

In preparing this manuscript, we employed a large language model (LLM) solely to assist with refining and polishing the text. The LLM was used to improve clarity, coherence, and readability, as well as to ensure consistent terminology throughout the paper. Importantly, all technical content, experimental design, and results were independently developed and verified by the authors; the LLM did not contribute to any scientific or methodological aspects of the work.

