# OpenReview forum: "Machine Unlearning under Retain–Forget Entanglement"
_ICLR.cc/2026/Conference — ICLR 2026 Poster_

### Official Review · Reviewer_TtJK · 2025-10-27

**Soundness:** 3
**Presentation:** 3
**Contribution:** 2
**Rating:** 4
**Confidence:** 3

**Summary:**

This paper focuses on machine unlearning, particularly focusing on the performance of the retain set samples that are highly related to the forget set. The forgetting is achieved in two steps. The first stage is very similar to the usual forgetting formulation to balance out retaining and forgetting, except that an adaptive weight is applied. The second stage uses a loss based on Wasserstein-2 to constrain the distributional change with respect to the forget set, which then forms a span for the retain set gradients to perform gradient projections.

**Strengths:**

The paper is written quite clearly.

Theoretical support for the paper seems to be substantial, and there is a bound on the accuracy of the forget set after PGD. Such theoretical support is good, although I did not check the derivations in too much detail.

**Weaknesses:**

Overall, while I get the motivation of trying to retain good performance on difficult samples, i.e., the samples in the retain set that are highly related to samples in the forget set, I am not quite convinced of the usefulness of such a setting. For instance, one needs to further define what are the “closely related” samples and pick some out for such further training, which seems somewhat arbitrary. Such picking of samples may be more straightforward in cases with clear classes (e.g., in CIFAR classification) and for class-level unlearning, but may be difficult to transfer to more general settings, e.g., when individual samples need to be forgotten.

Although the explanation for the gradient projection with Wasserstein distance regulation is mostly clear, I feel like some intuition is missing. For instance, what does it mean to project the adjacent set gradients onto the orthogonal complement of the space spanned by the forget and retain gradients? Furthermore, why must it be the span of those specific forget and retain gradients?


There are also some weaknesses in the experiments section, which have been placed in the “Questions” section.

**Questions:**

In the experiments, the compared methods, while representative, seem slightly outdated. For instance, SCRUB, SalUn and SSD were all available online already in 2023, and it has been more than two years. Also, the most recent among them was officially published in AAAI 2024, which is in Feb 2024.

At the same time, there seems to be an error in the citations for SCRUB:
Towards Unbounded Machine Unlearning is a NeurIPS 2023 paper, not 2024.


The authors can consider reporting results and/or comparing with more recent works such as:

Learning to Unlearn for Robust Machine Unlearning. ECCV 2024

Adversarial Machine Unlearning. ICLR 2025

Decoupled Distillation to Erase: A General Unlearning Method for Any Class-centric Tasks. CVPR 2025

LoTUS: Large-Scale Machine Unlearning with a Taste of Uncertainty. CVPR 2025

MUNBa: Machine Unlearning via Nash Bargaining. ICCV 2025


Often, machine unlearning works report the “upper bound” performance, which is the gold model: a complete retraining of the models only on the retain set. This shows us how far we are away from that upper bound. The authors should provide this.

Many works also report the “Unlearning Time”, which is the amount of time used for unlearning. This is important, since a complete retraining of the model would be ideal, but would consume too much resources. The authors should provide this.

---

> ### Author Response · Authors · 2025-11-24
>
> We thank the reviewer for the constructive comments.
>
> ### W1. Identification of adjacent and remote retain samples
>
> We kindly refer to our general comment to all reviewers on this point. Briefly, in all our experiments the adjacent subset is task-defined (subclass within superclass in vision; group-based semantics in ToxiGen), and our method is designed to improve the unlearning performance once such entanglement has been identified. However, the systematic discovery of adjacent retains is an important but orthogonal problem.
>
> ### W2. Intuition for the gradient projection and why we use the span of those specific gradients.
>
> We will substantially expand the intuition in Section 4.2.2. The key idea is that Stage 2 is solving a *local multi-objective problem*:
> - **Objective A:** decrease the loss on the adjacent retain set $\mathcal{D}_r^{\text{adj}}$;
> - **Objectives B & C:** avoid changing the modified forget loss $\tilde L_f$ and the remote-retain loss $L(\theta; \mathcal{D}_r^{\text{rem}})$.
>
> The gradients $\nabla_\theta \tilde L_f(\theta)$ and $\nabla_\theta L(\theta; \mathcal D_r^{\text{rem}}) $ capture the local directions along which the forget and remote objectives are most sensitive. By projecting $\nabla_\theta L(\theta; \mathcal{D}_r^{\text{adj}})$ onto the orthogonal complement of their span, we explicitly remove the components that would change these objectives to first order. Geometrically, this is the standard construction for maintaining constraints in a gradient step: we project the adjacent-retain gradient into the feasible tangent space defined by the two constraints.

---

> > ### Author Response · Authors · 2025-11-24
> >
> > ### Q1. Additional comparison to more recent methods.
> >
> > We sincerely thank this suggestion and agree that including recent methods strengthens the empirical validation.
> >
> > - We apologize for the mistake in the SCRUB citation. We have corrected it in the revised version.
> > - In the revision, we have already added comparisons to three recent methods whose code and settings align well with our scenario:
> >   - **DELETE** (CVPR 2025): a distillation-based class-centric unlearning method;
> >   - **GDR** (ACMMM 2024): a direction-rectified gradient method addressing gradient conflicts;
> >   - **MUNBa** (ICCV 2025): a Nash-bargaining based multi-objective unlearning method.
> >
> > The results are reported in section 5.2–5.5 across CIFAR-100 (Table 1), TinyImageNet (Table 2), and ToxiGen (Table 4) in our revised submission. For example, the comparison of our method with these three recent methods on CIFAR-100 superclass unlearning is as follows:
> >
> > | Method | Train $\mathcal D_f$ | Train $\mathcal D_r^{\text{adj}}$ | Train $\mathcal D_r^{\text{rem}}$ | Test $\mathcal D_f$ | Test $\mathcal D_r^{\text{adj}}$ | Test $\mathcal D_r^{\text{rem}}$ |
> > |---------|----------------------|----------------------------------|----------------------------------|----------------------|----------------------------------|----------------------------------|
> > | DELETE  | $0.00_{\pm 0.00}$   | $3.57_{\pm 0.18}$                | $98.37_{\pm 0.29}$               | $0.67_{\pm 0.47}$   | $2.83_{\pm 0.66}$                | $82.09_{\pm 0.37}$               |
> > | MUNBa   | $33.80_{\pm 8.88}$  | $92.17_{\pm 2.57}$               | $92.68_{\pm 1.28}$               | $31.67_{\pm 4.78}$  | $69.75_{\pm 3.74}$               | $75.32_{\pm 1.88}$               |
> > | GDR     | $4.87_{\pm 1.05}$   | $31.92_{\pm 6.45}$               | $96.10_{\pm 0.32}$               | $8.67_{\pm 1.25}$   | $22.33_{\pm 4.59}$               | $79.93_{\pm 0.09}$               |
> > | Ours    | $0.00_{\pm 0.00}$   | $98.17_{\pm 0.31}$               | $98.44_{\pm 0.05}$               | $2.33_{\pm 0.47}$   | $78.17_{\pm 0.31}$               | $81.10_{\pm 0.18}$               |
> >
> > DELETE and GDR both achieve strong forgetting (low $\mathcal D_f$ accuracy) but substantially degrade adjacent-retain performance, suggesting that the aggressive gradient manipulation or distillation process over-corrects nearby samples. MUNBa attains a better balance between forgetting and retention, yet still incurs a noticeable accuracy drop on $\mathcal D_r^{\text{adj}}$ and weaker forgetting compared to our method.
> > By contrast, our method achieves **near-zero forget accuracy** while preserving **high accuracy** on both adjacent and remote retain subsets. This shows that our method effectively mitigates retain–forget interference, outperforming these baselines in this entangled setting.

---

> > > ### Author Response · Authors · 2025-11-24
> > >
> > > ### Q2. Retraining as an upper bound
> > >
> > > We clarify that as we mentioned in line 138–144, our focus is on the goal of maximally reducing the model’s performance on the forget set, as proposed in [Karmanji et al., 2023], rather than achieving a close performance to the retrained model without the forget data. Therefore, in this setting, the retraining model is not necessarily the "upper bound". However, we provide the comparison of our method with the retraining one on ResNet-18 below, with the forget set being "aquarium fish".
> > >
> > > | Method    | $D_f^{\text{train}}$           | $D_r^{\text{in, train}}$        | $D_r^{\text{out, train}}$       | $D_f^{\text{test}}$            | $D_r^{\text{in, test}}$         | $D_r^{\text{out, test}}$        |
> > > |-----------|--------------------------------|----------------------------------|----------------------------------|--------------------------------|----------------------------------|----------------------------------|
> > > | Retrain   | $38.4_{\pm 3.80}$              | $99.98_{\pm 0.02}$              | $99.99_{\pm 0.00}$              | $37.00_{\pm 5.10}$             | $83.92_{\pm 4.20}$              | $83.37_{\pm 0.31}$              |
> > > | Our method| $0.00_{\pm 0.00}$              | $98.17_{\pm 0.31}$              | $98.44_{\pm 0.05}$              | $2.33_{\pm 0.08}$              | $78.17_{\pm 0.31}$              | $81.10_{\pm 0.18}$              |
> > >
> > > Compared to the retraining model, our method achieves significantly lower accuracy on the forget set. This indicates that the model has largely eliminated the influence of these samples.
> > >
> > > ### Q3. Report unlearning time.
> > >
> > > We have reported the wall-clock time and memory usage in our response to Reviewer LxMc (W3). Briefly, although our method introduces some additional computational time compared to certain baselines, this cost remains modest and significantly lower than complete retraining. We believe the time and memory overhead represent a favorable trade-off, as our primary contribution is the **improved unlearning performance**, with overhead that remains within acceptable limits.
> > >
> > > **References:**
> > > - Wu, Jing, and Mehrtash Harandi. "Munba: Machine unlearning via Nash bargaining." *ICCV 2025*.
> > > - Zhou, Yu, et al. "Decoupled distillation to erase: A general unlearning method for any class-centric tasks." *CVPR 2025*.
> > > - Lin, Shen, et al. "GDR-GMA: Machine Unlearning via Direction-Rectified and Magnitude-Adjusted Gradients." *ACM Multimedia 2024*.

---

### Official Review · Reviewer_SCFy · 2025-10-30

**Soundness:** 3
**Presentation:** 3
**Contribution:** 3
**Rating:** 6
**Confidence:** 4

**Summary:**

The paper proposes a two-stage optimization framework for correlation-aware unlearning. Stage 1 establishes forgetting while preserving performance on remote retain samples. Stage 2 recovers performance on adjacent retain samples while provably constraining the forget-set loss distribution via Wasserstein-distance regularization (W2). Experiments on CIFAR-100, Tiny-ImageNet, and ToxiGen across multiple model backbones demonstrate the effectiveness of the proposed method on both adjacent and remote retain subsets.

**Strengths:**

1. The problem is well defined, the experimental setup is sound, and the results of the proposed method are promising; the ablation studies are comprehensive.

2. For Stage 2, the analysis of why vanilla PGD fails is interesting and motivates the use of W2 regularization.

**Weaknesses:**

1. The theoretical components are doubtful, but Proposition 4.1—which relates the update to the learning rate and claims the modified loss remains largely unchanged—does not fully convince; stronger bounds or clearer assumptions would help establish this as evidence.

2. Notation could be clearer: prefer $L\left(\theta ; D_r^{\mathrm{adj}}\right)$ instead of $L_r^{\mathrm{adj}}(\theta)$, and likewise $L\left(\theta ; D_r^{\mathrm{rem}}\right)$ and $L\left(\theta ; D_f\right)$.

**Questions:**

1. Could you give more explanations about the failure of PGD?

2. What is $m$ in Proposition 4.2.

---

> ### Author Response · Authors · 2025-11-24
>
> We thank the reviewer for the positive assessment and valuable suggestions.
>
> ### W1. Clarification on the theoretical results
>
> Proposition 4.1 is intended as a local statement about one W-PGD update with a sufficiently small learning rate $\eta$: after projecting the adjacent-retain gradient onto the orthogonal complement of the span of the forget and remote gradients, the first-order change in the modified loss on the forget/remote sets vanishes, and only second-order ($\mathcal{O}(\eta^2)$) terms remain. If we have further regularity assumptions on the loss function, e.g. the boundedness of the Hessian, then the second-order upper bound can be further controlled. However, in general, the result provided in Proposition 4.1 is the best we can offer without additional assumptions.
>
> ---
>
> ### W2. Notation improvements
>
> We agree that this notation is clearer. In the revised version we make a clarification on the notation $L_r^{\text{adj}}(\theta)$ and $L_r^{\text{rem}}(\theta)$ when they are first introduced in section 4.1 (line 158).
>
> ### Q1. Additional explanation on the failure of PGD
>
> Intuitively, PGD constrains only the *mean* loss on the forget set. During Stage 2, the optimizer can keep this mean fixed by increasing the loss on a subset of harder forget examples while *decreasing* the loss on another subset. This “loss redistribution” leads to a skewed loss distribution: many forget samples end up with small loss (and thus high accuracy), even though the average loss has not changed.
>
> Concretely, Figure 1(b–c) shows that after PGD the loss histogram on $\mathcal{D}_f$ becomes highly polarized, with a non-trivial mass near zero loss. As a result, forget-set accuracy increases despite the mean loss being preserved.
> This motivates W-PGD: by adding a $W_2$ penalty on the loss *distribution* over $\mathcal{D}_f$, we restrict such redistributions and keep the entire loss histogram close to its post–Stage-1 shape, which empirically prevents the large distribution shift over the forget set.
>
> ---
>
> ### Q2. Explanation on $m$ in Proposition 4.2
>
> We apologize for the lack of clarity. In Proposition 4.2, $m$ is a *lower bound on the per-sample loss* on the forget set after Stage 1, i.e.
> $$
> \ell(f_{\bar\theta}(x_i), y_i) \geq m \quad \text{for all } (x_i, y_i) \in \mathcal{D}_f.
> $$
> The condition $m > \log n$ ensures that every forget sample is already assigned a large loss under $\bar\theta$ (the post–Stage-1 parameter), so that any sample that becomes correctly classified after Stage 2 must incur a substantial decrease in its loss. Combining this with the $W_2$ constraint on the loss distribution yields the stated upper bound on the accuracy $\mathrm{Acc}_f(\theta)$.
>
> In the revision we will restate Proposition 4.2 with this interpretation: "$m$ is a lower bound on losses over $\mathcal{D}_f$ after Stage 1.”

---

### Official Review · Reviewer_LxMc · 2025-10-31

**Soundness:** 3
**Presentation:** 2
**Contribution:** 3
**Rating:** 4
**Confidence:** 4

**Summary:**

The paper tackles retain–forget entanglement in machine unlearning: forgetting a targeted subset often harms performance on semantically adjacent retained examples. The authors formalize the retain set as two parts, adjacent (correlated with the forget set) and remote (less related), and propose a two-stage optimization framework (TMU). Stage 1 uses an augmented Lagrangian to increase loss on the forget set while constraining performance on the remote retain set. Stage 2 applies projected gradient descent on the adjacent retain set but augments the projection with a Wasserstein-2 (W2) loss-distribution regularizer so that improvements on adjacent retains do not "re-enable" the forget set (i.e., keep forget accuracy low by maintaining the post-Stage-1 loss distribution). The paper provides propositions showing that Stage 2 decreases adjacent-retain loss while keeping changes to the forget/remote objectives second-order, and derives an upper bound on forget-set accuracy under the W2 constraint. Experiments on CIFAR-100 (ResNet-18), Tiny-ImageNet (ViT), and ToxiGen (RoBERTa-base) show lower forget accuracy with competitive or better retain performance than baselines (Fine-tune, GA, SCRUB, SSD, SalUn, L1-sparse). Ablations indicate W2 regularization is key; membership-inference evaluation reports high "unlearning privacy" for the proposed method.

**Strengths:**

1. Explicitly decomposing the retain set into adjacent vs. remote is well-motivated and aligns with observed collateral damage patterns in unlearning.
2. Stage 1’s augmented-Lagrangian constraint is a clean way to preserve remote-retain performance without manual trade-off tuning; Stage 2’s W2-regularized projection addresses the empirical failure mode of plain PGD (mean-loss conservation but accuracy creep on the forget set).
3. Propositions articulate why adjacent-retain improves while forget/remote are (first-order) preserved; the accuracy bound connects the W2 constraint to forgetting fidelity.
4. Results span vision and language with multiple architectures; tables consistently show strong forget fidelity with limited adjacent-retain degradation, plus a clear W2 ablation and a privacy (MIA) check.

**Weaknesses:**

1. The method presumes a way to define and extract the adjacent subset (e.g., via superclasses or group semantics). Practical pipelines may not expose such structure; guidance or automated discovery (e.g., metric-learning neighbors) is not fully developed.
2. While datasets are standard, scenarios remain largely supervised classification with relatively clean adjacency (superclass/subclass or group semantics). It’s less clear how well the approach extends to open-world or multilabel settings, representation-level forgetting, or generative models.
3. Two stages (with $\lambda$, $\mu$, $\alpha$, step sizes, and projection subspaces) add complexity. The ablations help, but a deeper resource/efficiency analysis and stability study across wider seeds/tasks would strengthen claims of robustness.
4. The accuracy bound leverages conditions (e.g., sufficiently large losses post-Stage-1 and batchwise W2) that may not strictly hold in streaming or highly imbalanced forget sets; practical guidance for setting $\alpha$ and monitoring $\epsilon$ is limited.
5. A growing body of work frames unlearning as a multi-objective / multi-task optimization between forget and retain objectives, using gradient projection or surgery ([1], [2]), robustness-oriented or meta-learning variants ([3], [8]), parameter/connection sensitivity ([4]), bargaining/game-theoretic scalarization ([6]), and post-unlearning alignment or re-learning prevention in generative models ([5], [7], [9]). These methods surface well-known issues—gradient interference, oscillatory updates, sensitivity to weights in scalarization, and Pareto inefficiency—that your two-stage constrained-then-projected approach is implicitly designed to mitigate. However, the paper does not explicitly position TMU (proposed method) against this family or analyze when/why its W2-regularized Stage-2 projection outperforms (or fails relative to) gradient-rectified updates, Nash bargaining, or meta-unlearning schedules. A tighter comparative discussion, plus side-by-side experiments under a common protocol (same forget/retain splits, identical budgets), with diagnostics such as gradient-angle statistics, Pareto AUC, and "re-enablement" rates, would clarify the contribution boundary and strengthen claims over multi-task baselines.

[1] Learn to Unlearn for Deep Neural Networks: Minimizing Unlearning Interference with Gradient Projection, WACV 2024.

[2] GDR-GMA: Machine Unlearning via Direction-Rectified and Magnitude-Adjusted Gradients, ACMMM 2024.

[3] Learning to Unlearn for Robust Machine Unlearning, ECCV 2024.

[4] Scissorhands: Scrub Data Influence via Connection Sensitivity in Networks, ECCV 2024.

[5] Boosting alignment for post-unlearning text-to-image generative models, NeurIPS 2024.

[6] MUNBa: Machine Unlearning via Nash Bargaining, ICCV 2025.

[7] Meta-Unlearning on Diffusion Models: Preventing Relearning Unlearned Concepts, ICCV 2025.

[8] Learning to Unlearn while Retaining: Combating Gradient Conflicts in Machine Unlearning, ICCV 2025.

[9] GRU: Mitigating the Trade-off between Unlearning and Retention for LLMs, ICML 2025

**Questions:**

1.  How robust is TMU to noisy adjacency (e.g., approximate neighbors via embeddings)? A study where adjacency is inferred (not oracle) would boost practical relevance.
2. Provide a simple recipe for setting $\alpha$ and stopping criteria for Stage 2 (e.g., target W2 window + retain gains plateau) to avoid overfitting adjacent retains.
3. Report wall-clock and memory overhead vs. SalUn/SSD/SCRUB on larger backbones and longer runs; clarify sorting cost for empirical W2 and any minibatch bias.

The paper identifies an important practical failure mode (retain–forget entanglement), proposes a simple-yet-effective two-stage remedy grounded in standard optimization plus a neat W2 distributional guardrail, and backs it with solid empirical evidence and helpful theory. The main asks are around adjacent-set construction, scalability/sensitivity details, and broader realistic applicable scenarios. Addressing these would elevate the contribution further.

---

> ### Author Response · Authors · 2025-11-24
>
> We thank Reviewer 2 for the constructive comments and suggestions.
>
> ### W1. Identification of adjacent and remote retain samples
>
> Please refer to our general comment to all reviewers.
>
> Briefly, we stress that adjacency is *task-defined* in our experiments (superclass/subclass structure in vision; group semantics in ToxiGen), and that our method is designed to improve unlearning once such a subset is specified. We additionally show robustness to noisy adjacency through a dedicated experiment (see Q1 below).
>
> ### W2. Generalizability to other unlearning scenarios
>
> Our experiments focus on supervised classification tasks because retain–forget entanglement is most interpretable and measurable in this setting (via class-wise accuracies and loss distributions). However, our framework itself is not tied to classification.
>
> Our approach essentially requires only:
>
> 1. A **task loss** $L(\theta; \cdot)$ defined on the forget, adjacent, and remote subsets, and
> 2. A **distributional discrepancy measure** (here, $W_2$ distance on scalar losses) to constrain Stage-2 updates.
>
> Thus, in principle, the same two-stage design—constrained forgetting on $\mathcal D_f$ and $\mathcal D_r^{\text{rem}}$, then $W_2$-regularized gradient projection to recover performance on $\mathcal D_r^{\text{adj}}$—can be generalized to other unlearning scenarios. We refer to our new CelebA experiment in response to Reviewer 1 as an exploration in the multi-label setting. Although the task is still instantiated as a 4-way single-label classification, the classes are defined as combinations of multiple underlying binary attributes (Male, Smiling, Young, Eyeglasses). This setting already reflects a richer multi-attribute structure than standard one-hot class labels.
>
> For generative models, we note that $W_2$ on scalar cross-entropy losses is no longer the most natural choice; instead, one may need some surrogate loss to penalize the distributional shift, which is an interesting but non-trivial extension and beyond the scope of this paper.
>
> ### W3. Resource/efficiency results and stability to hyperparameters
>
> - **Wall-clock time.** In Appendix B.3 of our submission, we report basic runtime metrics on CIFAR-100 with ResNet-18. The result shows that our method incurs only a modest runtime overhead relative to SalUn/SSD/SCRUB (on the order of 10–15% additional training time).
> - **Memory.** We report the memory usage in the following table, where all the methods use the same batch size of 128.
> The results show that our method does not introduce significant memory overhead compared to the baselines. While our method does introduce some additional computational time compared to certain baselines, this cost remains modest and significantly lower than complete retraining.
>
> | Method       | Retrain | FT   | GA   | SCRUB | $\ell_1$-sparse | SalUn | Ours |
> |--------------|---------|------|------|-------|------------------|-------|------|
> | Memory (MB)  | 2949    | 2949 | 2860 | 3715  | 2974             | 4993  | 3297 |
>
> We believe the time and memory overhead represent a favorable trade-off, as our primary contribution is the **improved unlearning performance**, with overhead that remains within acceptable limits.
>
> - **Hyperparameters.** We first clarify that $\lambda$ is not essentially a hyperparameter, as it is initialized as zero and automatically adjusted via the constraints and the hyperparameter $\mu$. For the hyperparameter $\alpha$, we have included additional discussion.
> We provide an additional sensitivity study on $\mu$ showing that the result is stable when $\mu$ varies within a reasonable range.
>
> | $\mu$ | Train $\mathcal D_f$ | Train $\mathcal D_r^{\text{adj}}$ | Train $\mathcal D_r^{\text{rem}}$ | Test $\mathcal D_f$ | Test $\mathcal D_r^{\text{adj}}$ | Test $\mathcal D_r^{\text{rem}}$ |
> |:-----:|-----------------------|-----------------------------------|-----------------------------------|---------------------|-----------------------------------|-----------------------------------|
> | 5     | $0.00_{\pm 0.00}$     | $98.00_{\pm 0.05}$               | $98.33_{\pm 0.02}$               | $3.00_{\pm 0.00}$  | $77.83_{\pm 0.52}$               | $81.08_{\pm 0.11}$               |
> | 10    | $0.00_{\pm 0.00}$     | $98.17_{\pm 0.31}$               | $98.44_{\pm 0.05}$               | $2.33_{\pm 0.47}$  | $78.17_{\pm 0.31}$               | $81.10_{\pm 0.18}$               |
> | 20    | $0.00_{\pm 0.00}$     | $98.17_{\pm 0.06}$               | $98.45_{\pm 0.12}$               | $2.33_{\pm 0.58}$  | $78.17_{\pm 0.63}$               | $80.97_{\pm 0.05}$               |
>
> These results support our claim that our method is computationally practical and robust to hyperparameter choices.

---

> > ### Author Response · Authors · 2025-11-24
> >
> > ### W4. On theoretical bounds and practical guidance for choosing $\alpha$
> >
> > Our accuracy bound (Prop. 4.2) is designed to characterize when the forgetting on $\mathcal{D}_f$ is guaranteed to be strong after Stage 2. The requirement that the losses on $\mathcal{D}_f$ are “sufficiently large” after Stage 1 is not an extra unrealistic assumption but exactly the optimization objective of Stage 1: the augmented Lagrangian is explicitly trained to drive up the loss on $\mathcal{D}_f$ while preserving $\mathcal{D}_r^{\text{rem}}$. Once this objective is achieved, the conditions of the bound are naturally met. Importantly, the bound is independent of the dataset size and hence continues to apply in imbalanced regimes, as long as Stage 1 achieves effective forgetting on the (possibly small) forget set.
> >
> > Regarding the choice of $\alpha$, in our experiments we observe that any $\alpha \ge 0.5$ already yields strong forgetting, and performance is not very sensitive within this range. The parameter $\varepsilon$ measures how much the $W_2$-regularized loss can change during Stage 2; with a reasonably small learning rate for the projected updates, this deviation remains small in practice. Explicitly monitoring $\varepsilon$ at each step would introduce non-trivial extra computation while offering limited practical benefit. In applications it is more direct and informative to monitor the empirical accuracy (and loss) on $\mathcal{D}_f$ itself, which is what practitioners ultimately care about. Thus, the primary role of our bound is not as a tuning rule, but as a theoretical justification that, under the training regime we use, the $W_2$-regularized projection step indeed preserves forgetting on $\mathcal{D}_f$ while improving performance on the adjacent retain set.
> >
> > ### W5. Further comparative discussion to other works
> >
> > We sincerely thank the detailed list of related works. Many of them can be viewed as multi-objective or multi-task unlearning methods, balancing forget and retain objectives via gradient projection/surgery, rectified gradients, bargaining, or meta-learning schedules.
> >
> > Ours is closely related but adds a **key new ingredient**: *distribution-level control* over the forget-set losses.
> >
> > - Existing projection-based methods typically ensure that the **mean loss** of certain objectives is controlled or maximized/minimized along updates, which can still allow **loss redistribution** within the forget set.
> > - As we show in Figure 1 of the submission, this mean-level control alone is insufficient, especially in the retain–forget entanglement setting: the forget-set loss distribution becomes highly skewed under PGD, with many samples drifting back to low loss and thus high accuracy, even when the mean loss is stable.
> > - TMU augments this by constraining the **$W_2$ distance between loss distributions** before and after Stage 2, and Proposition 4.2 translates this into an explicit bound on forget accuracy.
> >
> > In the revision, we include additional comparison with three recent methods (DELETE, GDR, MUNBa) across CIFAR-100, TinyImageNet, and ToxiGen. The results indicate that these methods generally underperform relative to ours in the presence of retain–forget entanglement, further validating the importance of distribution-level control.

---

> > > ### Author Response · Authors · 2025-11-24
> > >
> > > ### Q1. How robust is TMU to noisy adjacency?
> > >
> > > We kindly refer to our general comment to all reviewers on this point. Briefly, we conducted a robustness study on noisy adjacency in the CIFAR-100 superclass unlearning task. The results demonstrate that our method remains robust.
> > >
> > > ### Q2. Recipe for setting $\alpha$ and stopping criteria for Stage 2
> > >
> > > - We set $\alpha=0.5$ across all experiments. In general, setting $\alpha \ge 0.5$ yields a good forgetting performance. See our response to W4 above for more discussion.
> > > - For stopping Stage 2, we provide the following training dynamics for Stage 2 in the CIFAR-100 superclass unlearning task. Although there is a slight performance drop on the remote retain set during training, we did not observe significant overfitting. In practice, the stopping criterion can be simply based on whether the accuracy on retain sets achieves a satisfactory level.
> > >
> > > | Metric | Epoch 0 | Epoch 1 | Epoch 2 | Epoch 3 | Epoch 4 | Epoch 5 | Epoch 6 | Epoch 7 | Epoch 8 | Epoch 9 | Epoch 10 |
> > > |---------|---------|---------|---------|---------|---------|---------|---------|---------|---------|---------|----------|
> > > | $\mathcal D_r^{\text{adj}}$ Acc (%) | 100.00 | 99.35 | 98.60 | 98.48 | 98.39 | 98.33 | 98.24 | 98.21 | 98.17 | 98.19 | 98.14 |
> > > | $\mathcal D_r^{\text{rem}}$ Acc (%) | 0.10 | 46.75 | 85.15 | 93.05 | 96.05 | 97.20 | 97.85 | 98.45 | 98.85 | 99.50 | 99.70 |
> > > | $\mathcal D_f$ Acc (%) | 0.00 | 0.00 | 0.00 | 0.00 | 0.00 | 0.00 | 0.00 | 0.00 | 0.00 | 0.00 | 0.00 |
> > >
> > > ### Q3. Wall-clock and memory overhead; sorting cost and bias for $W_2$ computation
> > >
> > > See our response to W2 above for wall-clock and memory overhead.
> > >
> > > For the $W_2$ computation, the sorting operation is just a standard sorting of an array of size equal to the batch size (128 in our experiments). The computational cost is negligible compared to the forward and backward passes of the neural network.
> > >
> > > We agree that there can be mini-batch bias when estimating $W_2$ from mini-batch losses. However, there is an inequality showing that $E[X-Y] \le W_2(P,Q)$ for any random variables $X\sim P$ and $Y\sim Q$. Taking $P$ and $Q$ to be the mini-batch empirical loss distributions, we know that even in the mini-batch setting, the $W_2$ distance is a stronger control than just punishing the mean loss difference. Moreover, our experiments empirically show that even with mini-batch estimation, our method effectively prevents the forget loss distribution from collapsing back to low-loss patterns.
> > >
> > > **References:**
> > > - Wu, Jing, and Mehrtash Harandi. "Munba: Machine unlearning via Nash bargaining." *ICCV 2025*.
> > > - Zhou, Yu, et al. "Decoupled distillation to erase: A general unlearning method for any class-centric tasks." *CVPR 2025*.
> > > - Lin, Shen, et al. "GDR-GMA: Machine Unlearning via Direction-Rectified and Magnitude-Adjusted Gradients." *ACM Multimedia 2024*.

---

> > > > ### Comment · Reviewer_LxMc · 2025-11-26
> > > > **Thanks for the rebuttal!!**
> > > >
> > > > I would like to thank the authors for their response in clarifying some of the concerns raised, however, I would like to maintain my original assessment score “marginally below acceptance threshold.”
> > > >
> > > > While TMU (proposed method) offers a strong motivation, the construction of the adjacent set remains an externally assumed component, and none of the practical, e.g., representation-driven strategies for discovering adjacency are incorporated or evaluated (k-NN in representation space, clustering, or influence-based metrics) in constructing the adjacent set. In real-world settings, noise in adjacency estimation, in non-ideal datasets that do not shave structured hierarchy, will compound and can significantly affect unlearning fidelity; hence, the current method’s robustness to such dataset partitions should be evaluated. Moreover, I second Reviewer TtJK’s point: the retrain baseline remains the fundamental gold standard for unlearning, especially for classification. Even if its forget accuracy is imperfect, a direct comparison to the retrain oracle is essential in understanding how well the proposed unlearning methods mimic the oracle [1,2,3,4].
> > > >
> > > > [1] Model Sparsity Can Simplify Machine Unlearning, NeurIPS 2023
> > > > [2] SalUn: Empowering Machine Unlearning via Gradient-based Weight Saliency in Both Image Classification and Generation, ICLR 2024
> > > > [3] Scissorhands: Scrub Data Influence via Connection Sensitivity in Networks, ECCV 2024
> > > > [4] MUNBa: Machine Unlearning via Nash Bargaining, ICCV 2025

---

> > > > > ### Author Response · Authors · 2025-11-29
> > > > >
> > > > > We thank the reviewer for the follow-up comments and provide additional clarification on the points raised.
> > > > >
> > > > > ### kNN-based identification of the adjacent retain set
> > > > >
> > > > > To further address the concern about how the adjacent retain set is identified, we additionally test our method on CIFAR-100 using a **feature-space kNN criterion** instead of task-defined superclasses.
> > > > >
> > > > > Concretely, we extract output features from the pretrained ResNet-18, compute the $k$ nearest neighbors ($k=20$) of each forget sample among all retained samples, and assign each retained point an adjacency score equal to how often it appears in these kNN lists. The top 10% by this score are treated as the **kNN adjacent retain set**, and the rest as the **kNN remote retain set**, on which we then run our two-stage method.
> > > > >
> > > > > The results below show that, under this automatically discovered adjacency, our method continues to achieve strong forgetting while maintaining high retain accuracy, and overall performance remains comparable to the task-defined setting.
> > > > >
> > > > > | Method                     | Train $D_f$           | Train $D_r^{\text{adj}}$ | Train $D_r^{\text{rem}}$ | Test $D_f$            | Test $D_r^{\text{adj}}$ | Test $D_r^{\text{rem}}$ |
> > > > > |----------------------------|------------------------|---------------------------|---------------------------|------------------------|--------------------------|--------------------------|
> > > > > | Ours (task-defined)        | $0.00_{\pm 0.00}$      | $98.17_{\pm 0.31}$        | $98.44_{\pm 0.05}$        | $2.33_{\pm 0.47}$      | $78.17_{\pm 0.31}$       | $81.10_{\pm 0.18}$       |
> > > > > | Ours (kNN-identified)      | $3.00_{\pm 0.75}$      | $99.31_{\pm 0.31}$        | $99.85_{\pm 0.07}$        | $6.00_{\pm 0.82}$      | $77.67_{\pm 0.31}$       | $83.13_{\pm 0.23}$       |
> > > > >
> > > > > We have also updated this in Appendix B.6 in the revised manuscript.
> > > > >
> > > > >
> > > > > ### Additional comparison with retraining
> > > > >
> > > > > In response to the question of how our method compares to full retraining, we additionally report retrain baselines on CIFAR-100, TinyImageNet, and ToxiGen under the *same forget/retain splits* as in our experiments. As shown below, retraining does not necessarily achieve strong erasure on the forget set in our setting.
> > > > >
> > > > > | Dataset       | Method    | Train $D_f$                 | Train $D_r^{\text{adj}}$         | Train $D_r^{\text{rem}}$         | Test $D_f$                  | Test $D_r^{\text{adj}}$          | Test $D_r^{\text{rem}}$          |
> > > > > |--------------|-----------|-----------------------------|----------------------------------|----------------------------------|-----------------------------|----------------------------------|----------------------------------|
> > > > > | CIFAR-100    | Retrain   | $38.40_{\pm 3.80}$          | $99.98_{\pm 0.02}$               | $99.99_{\pm 0.00}$               | $37.00_{\pm 5.10}$          | $83.92_{\pm 4.20}$               | $83.37_{\pm 0.31}$               |
> > > > > | CIFAR-100    | Ours      | $0.00_{\pm 0.00}$      | $98.17_{\pm 0.31}$        | $98.44_{\pm 0.05}$        | $2.33_{\pm 0.47}$      | $78.17_{\pm 0.31}$       | $81.10_{\pm 0.18}$       |
> > > > > | TinyImageNet | Retrain   | $62.82_{\pm 5.59}$          | $99.46_{\pm 0.63}$               | $97.99_{\pm 2.01}$               | $64.45_{\pm 4.79}$          | $95.71_{\pm 0.23}$               | $90.58_{\pm 0.12}$               |
> > > > > | TinyImageNet | Ours      | $0.00_{\pm 0.00}$       | $98.95_{\pm 0.08}$        | $98.49_{\pm 0.09}$        | $3.11_{\pm 0.31}$        | $91.27_{\pm 0.78}$        | $88.88_{\pm 0.54}$        |
> > > > > | ToxiGen      | Retrain   | $8.58_{\pm 0.66}$           | $93.71_{\pm 2.08}$               | $91.50_{\pm 1.44}$               | $12.13_{\pm 4.34}$          | $91.74_{\pm 2.45}$               | $89.35_{\pm 2.73}$               |
> > > > > | ToxiGen      | Ours      | $11.95_{\pm 0.02}$      | $88.88_{\pm 0.01}$        | $92.73_{\pm 0.01}$        | $14.29_{\pm 0.06}$       | $85.86_{\pm 0.00}$        | $85.23_{\pm 0.01}$        |
> > > > >
> > > > >
> > > > > We have also updated the retrain result in Appendix B.6 in our revised manuscripts.

---

### Official Review · Reviewer_3FyN · 2025-11-04

**Soundness:** 3
**Presentation:** 3
**Contribution:** 3
**Rating:** 8
**Confidence:** 4

**Summary:**

This paper addresses retain–forget entanglement in machine unlearning, where removing a forget set unintentionally harms correlated retained data. It introduces a two-stage framework: (1) an augmented Lagrangian step to forget targeted data while preserving unrelated samples, and (2) a Wasserstein-regularized gradient projection to recover performance on correlated retained data. Experiments on CIFAR-100, TinyImageNet, and ToxiGen show that the method achieves strong forgetting while maintaining retention accuracy, outperforming prior baselines.

**Strengths:**

1. The paper pinpoints the practical challenge that forgetting one subset of data (e.g., a specific subclass or biased group) can harm semantically related retained samples, a scenario highly relevant for real-world fairness and bias correction.

2. Ablation studies clarify the importance of the Wasserstein-2 term.

3. The experimental design is well-defined and comprehensive, matching the problem setup.

**Weaknesses:**

1. The method relies on manual partitioning of the retained set into “adjacent” and “remote” subsets, whether it assumes prior knowledge of semantic relationships is unknown; If so, this decomposition may be impractical or ambiguous in large, unstructured datasets.

2. The experimental evaluation on vision task is limited to small scale datasets.

3. I did not get the meaning of theoretical bounds.

**Questions:**

How is the adjacent vs. remote retain set identified in practice for complex or unlabeled datasets?

---

> ### Author Response · Authors · 2025-11-24
>
> We thank Reviewer 1 for the thoughtful comments and positive evaluation.
>
> ### W1. Identification of adjacent and remote retain samples
>
> Please refer to our general comments to all reviewers on the identification of $\mathcal D_r^{\text{adj}}$ and $\mathcal D_r^{\text{rem}}$.
> In short, in all our experiments the adjacent subset is **task-defined** (subclass within superclass in vision; group-based semantics in ToxiGen), and our method is designed as a *mechanism* that improves unlearning once such entanglement has been identified. However, the systematic discovery of adjacent retains is an important but orthogonal problem.
>
> ### W2. The experimental evaluation on vision tasks is limited to small-scale datasets
>
> We agree that evaluating on a larger-scale vision task is valuable. In the revision, we add a new experiment on **CelebA**, a popular large-scale face attributes dataset containing over 200K images with 40 binary attribute annotations. The base model is ViT-B/32.
> Specifically, we construct a 4-class attribute-based classification task using the two binary attributes **Male** and **Smiling**.
> We then define the **forget set** as images from the “female & not smiling” class that are also **not Young** and **do not wear Eyeglasses**, and treat the remaining classes as retains (with those sharing similar attributes naturally forming the adjacent retain set). We report unlearning results for this setting in the following table and the revised manuscript.
>
> Fine-tuning, $\ell_1$-sparse, and SCRUB largely preserve accuracy on both retain subsets, but only achieve modest forgetting: test accuracy on $\mathcal{D}_f$ remains above $70\%$. Gradient Ascent and DELETE, on the other hand, drive the forget accuracy to essentially zero, but do so by collapsing performance on the adjacent retain set to chance level, rendering the model unusable on the very samples we aim to protect. SSD also degrades both adjacent and remote retain accuracy substantially.
> In contrast, our method achieves a significantly lower test accuracy on the forget set (from $81.37\%$ down to $25.48\%$) while still maintaining high accuracy on $\mathcal{D}_r^{\text{adj}}$ ($75.05\%$) and $\mathcal{D}_r^{\text{rem}}$ ($92.38\%$), yielding the best overall balance between effective forgetting and retention in this more demanding scenario. We notice that this task is more challenging than the CIFAR100 and TinyImageNet task. It is possibly due to that the subclasses in different superclasses can also be highly correlated, as there are essentially only 4 different attributes.
>
> | Method                     | Train $\mathcal D_f$              | Train $\mathcal D_r^{\text{adj}} $           | Train $\mathcal D_r^{\text{rem}} $            | Test $\mathcal D_f$               | Test $\mathcal D_r^{\text{adj}} $            | Test $\mathcal D_r^{\text{rem}} $             |
> |-----------------------------|------------------------|------------------------|------------------------|------------------------|------------------------|------------------------|
> | Origin            | $98.91$      | $99.08$     | $99.53$     | $81.37$      | $89.93$     | $90.82$     |
> | FT | $69.23_{\pm 6.86}$      | $89.97_{\pm 1.93}$     | $91.57_{\pm 2.70}$     | $67.56_{\pm 7.56}$      | $88.71_{\pm 2.89}$     | $89.77_{\pm 7.11}$     |
> | $\ell_1$-sparse | $76.03_{\pm 3.41}$      | $92.02_{\pm 1.73}$     | $90.48_{\pm 0.65}$     | $75.28_{\pm 4.42}$      | $91.87_{\pm 1.94}$     | $89.46_{\pm 0.76}$     |
> | GA | $0.00_{\pm 0.00}$      | $0.00_{\pm 0.00}$     | $97.46_{\pm 0.41}$     | $0.00_{\pm 0.00}$      | $0.00_{\pm 0.00}$     | $91.16_{\pm 0.46}$     |
> | SCRUB | $80.73_{\pm 3.25}$      | $96.81_{\pm 1.82}$     | $98.38_{\pm 0.64}$     | $71.1_{\pm 2.87}$      | $89.22_{\pm 2.11}$     | $90.57_{\pm 0.76}$     |
> | SSD | $23.07_{\pm 0.00}$      | $41.81_{\pm 0.00}$     | $84.28_{\pm 0.00}$     | $23.19_{\pm 0.00}$      | $44.52_{\pm 0.00}$     | $80.05_{\pm 0.00}$     |
> | DELETE | $0.00_{\pm 0.00}$      | $0.00_{\pm 0.00}$     | $99.62_{\pm 0.00}$     | $0.00_{\pm 0.00}$      | $0.00_{\pm 0.00}$     | $93.97_{\pm 0.01}$     |
> | Ours | $1.85_{\pm 0.09}$      | $85.65_{\pm 0.25}$     | $99.08_{\pm 0.42}$     | $25.48_{\pm 0.56}$      | $75.05_{\pm 0.34}$     | $92.38_{\pm 0.06}$     |

---

> > ### Author Response · Authors · 2025-11-24
> >
> > ### W3. Further explanations on the theoretical bounds
> >
> > Informally, Proposition 4.2 states that if, after Stage 1, the loss on the forget set $\mathcal D_f$ is uniformly large (i.e., there exists a lower bound $m$ on all per-sample losses), and if Stage 2 keeps both (i) the mean loss and (ii) the $W_2$ distance between the loss distributions (before vs. after Stage 2) sufficiently small, then the accuracy on $\mathcal D_f$ must also remain small. In other words, **once Stage 1 has pushed the forget set into a high-loss region, the W₂-regularized Stage-2 updates cannot “silently” bring back many correctly classified forget samples without paying a substantial $W_2$ penalty**.
> >
> > The intuition is as follows. Suppose after Stage 1 every forget sample has loss at least $m$, which means the model is confidently incorrect or at least far from the correct label on $\mathcal D_f$. If, during Stage 2, a non-negligible fraction of these samples were to become correctly classified, their losses would have to move from around $m$ down to values near 0. This would create a “mass shift” in the loss distribution on $\mathcal D_f$: a block of probability mass moves a distance of order $m$ along the loss axis. Such a shift necessarily induces a large $W_2$ distance between the pre–Stage-2 and post–Stage-2 loss distributions. Therefore, when we add a $W_2$ regularization term with a non-negligible weight $\alpha$, any update that significantly improves accuracy on $\mathcal D_f$ becomes very costly in the objective and is discouraged by the optimizer. The bound in Proposition 4.2 formalizes this intuition: **large post–Stage-1 losses plus a small W₂ change imply a small fraction of low-loss (hence correctly classified) forget samples, i.e., low forget accuracy**.
> >
> > We hope this explanation helps you understand. If you have further questions, we are happy to clarify.

---

### Author Response · Authors · 2025-11-24

### Clarification on identifying the adjacent retain set

We noticed that many reviewers raised related questions about how to *identify* the adjacent retain subset $\mathcal D_r^{\text{adj}}$. We would like to provide a general clarification on this point.

### 1. Scope of our paper
We authors would like to clarify that our **primary goal** of this paper is **not** to solve the discovery problem of “which retained samples are entangled with the forget set,” but to answer a different question:

- Given that such retain–forget entanglement exists (and that some adjacent subset can be specified), what **algorithm** should we use so that we can forget $\mathcal D_f$ while preserving $\mathcal D_r^{\text{adj}}$?

In the task we focus on in this paper, $\mathcal D_r^{\text{adj}}$ is part of the task specification rather than an unknown object: subclasses within the same superclass (CIFAR-100, TinyImageNet), sentences about the same social group (ToxiGen), or faces sharing similar attributes (CelebA).
In these cases, the adjacent retain set can be clearly and naturally defined based on task semantics. Our contribution is to **design and analyze an unlearning algorithm that explicitly protects this adjacent subset**, rather than proposing a new mechanism to discover it from scratch.

### 2. Existing work on identifying influential / adjacent samples
We also acknowledge that in many practical scenarios, the adjacent retain set may not be explicitly defined by task structure. We agree that automatically discovering which retained samples are most affected by unlearning is an important and complementary research direction, especially in more complex settings where task-defined adjacency is unavailable or insufficient.

For example, [Chang & Lee, 2025] systematically analyze how different retain subsets are affected during unlearning, and identify a syntactically similar neighbor set that suffers the largest utility drop.

[Niu et al., 2025] introduced a lightweight proxy data attribution metric tailored to LLMs, which quantifies the “alignment” between the Forget and Retain sets. These works are complementary to ours: any procedure that outputs a set of “highly influential” or “syntactically similar” retained samples can be used to instantiate our $\mathcal D_r^{\text{adj}}$.

### 3. Robustness to imperfect adjacency
To address the concern that $\mathcal D_r^{\text{adj}}$ may be only approximately specified in practice, we conduct a robustness study on the CIFAR-100 superclass unlearning task. Specifically, starting from the clean partition of the retain set into adjacent and remote subsets, we consider two noisy scenarios:

1. **Case 1:** 20% of random samples from the remote retain set are mis-identified as adjacent;
2. **Case 2:** 20% of random samples from the true adjacent retain set are mis-identified as remote.

The results are reported in the following table. Our method remains robust under such noisy adjacency: forget accuracy stays zero on training data, and also very low in test data, and the changes in adjacent/remote retain accuracy are modest. This suggests that our method does not require perfectly identified adjacency to be effective, and can tolerate reasonable noise in the partition.

| Setting                      | Train $\mathcal D_f$ | Train $\mathcal D_r^{\text{adj}}$ | Train $\mathcal D_r^{\text{rem}}$ | Test $\mathcal D_f$ | Test $\mathcal D_r^{\text{adj}}$ | Test $\mathcal D_r^{\text{rem}}$ |
|------------------------------|----------------------|-----------------------------------|-----------------------------------|----------------------|-----------------------------------|-----------------------------------|
| Clean adjacency              | $0.00_{\pm 0.00}$    | $98.17_{\pm 0.31}$               | $98.44_{\pm 0.05}$               | $2.33_{\pm 0.47}$   | $78.17_{\pm 0.31}$               | $81.10_{\pm 0.18}$               |
| + 20% non-adj → adj (case 1) | $0.00_{\pm 0.00}$    | $98.81_{\pm 0.20}$               | $98.40_{\pm 0.20}$               | $5.33_{\pm 0.58}$   | $81.37_{\pm 0.33}$               | $78.92_{\pm 0.95}$               |
| + 20% adj → non-adj (case 2) | $0.00_{\pm 0.00}$    | $93.93_{\pm 0.67}$               | $95.75_{\pm 0.31}$               | $5.00_{\pm 1.00}$   | $77.32_{\pm 0.41}$               | $80.08_{\pm 1.18}$               |

**References:**
- Chang, Hwan, and Hwanhee Lee. "Which retain set matters for LLM unlearning? A case study on entity unlearning." *arXiv preprint arXiv:2502.11441* (2025).
- Niu, Peizhi, et al. "Guard: Guided unlearning and retention via data attribution for large language models." *arXiv preprint arXiv:2506.10946* (2025).

---

### Meta-Review · Area_Chair_ZKRr · 2026-01-05

**Summary:**

The reviewers raised several reasonable concerns. The strongest and most recurring issue was the reliance on splitting the retain set into "adjacent" and "remote" subsets. Multiple reviewers questioned how this split would be identified in practice, especially in unstructured or real-world settings without clear semantic hierarchies. The authors acknowledged this limitation and clarified that adjacency is task-defined in their current experiments. While this does not fully resolve the concern, the authors strengthened their response by adding an automatic kNN-based adjacency construction and showing that the method remains effective under this noisier, inferred partition. This reduces, but does not eliminate, doubts about general applicability.

Other major concerns focused on understanding and justification rather than correctness. Several reviewers asked for clearer intuition on why vanilla projected gradient descent fails and why the Wasserstein2 constraint is necessary. The rebuttal provided an explanation: PGD preserves only the mean forget loss and allows loss redistribution, which can silently restore accuracy on forgotten samples. The authors' explanation of this failure mode, together with the W2-based distributional constraint, addresses this point. Related theoretical questions about the propositions and bounds were also clarified at a level appropriate for a rebuttal, even if the theory remains somewhat local and assumption-dependent.

Reviewers also requested broader and stronger empirical validation. These concerns were mostly addressed by adding experiments on a larger dataset (CelebA), incorporating comparisons with more recent unlearning methods, correcting citation issues, reporting retraining baselines, and providing runtime and memory overhead measurements. While the evaluation is still centered on supervised classification and does not cover all possible unlearning scenarios, the additional experiments and comparisons meaningfully strengthen the empirical case.

Overall, some limitations remain, but the core technical questions, experimental gaps, and clarity issues raised by reviewers were addressed reasonably well.

**Reviewer Concerns:**

Across reviewers, the rebuttal addressed most of the concrete technical and experimental requests. Multiple reviewers asked for clearer intuition on why vanilla PGD fails and what the projection is doing (SCFy, TtJK). The authors gave a partial but accepted explanation. This directly motivates the Stage 2 process so it is important. Questions about the theory were also largely handled at the level reviewers were asking for: they clarified Proposition 4.1 as a local, small-step, first-order argument (forget/remote objectives change only at second order after projection), and they clarified the meaning of the key term in Proposition 4.2 (a lower bound on per-sample forget loss after Stage 1) with an intuitive "mass shift" argument for why W2 penalizes relearning. Notation/citation issues were also acknowledged and corrected.

The rebuttal also improved the empirical side in ways reviewers explicitly requested. Concerns about small-scale vision evaluation (3FyN) were addressed by adding a CelebA experiment with a larger model (ViT-B/32) and a more attribute-driven setting. Concerns about outdated baselines and missing comparisons (TtJK) were addressed by adding newer methods and fixing the SCRUB citation. Concerns about resource overhead and stability (LxMc) were addressed by reporting basic wall-clock overhead (stated as modest, ~10–15% on CIFAR-100) and a memory table, plus a sensitivity study for key hyperparameters. The "retrain baseline" request was also addressed in the sense that the authors added retraining results (later reported across CIFAR-100, TinyImageNet, and ToxiGen) and discussed how their objective differs from simply matching the retrained model.

What remains outstanding is mainly the practicality and generality of the "adjacent vs. remote" split. This was the core weakness flagged by 3FyN, LxMc, and TtJK: the method depends on a decomposition that is easy in curated benchmarks (superclass/subclass or group semantics) but is ambiguous in messy real-world datasets. The authors initially treated discovery of adjacency as "orthogonal," which did not fully satisfy the concern. They later strengthened the rebuttal by adding a kNN-in-feature-space adjacency construction on CIFAR-100 and showing the method still works under that automatically inferred split. That's a meaningful step, but it's still limited. Similarly, the "retrain is gold standard" point is only partially resolved: the authors did add retrain comparisons, but their framing that retrain is not necessarily an "upper bound" may still not convince readers who view indistinguishability from retrain as the canonical goal in classification unlearning.

**Reviewer Scores:**

Reviewer 3FyN was already strongly positive (score 8) and did not raise fundamental objections. Their main questions were about how adjacent vs. remote retain sets are identified, the limited scale of vision experiments, and the meaning of the theoretical bounds. The rebuttal addressed most of them. Given this, I do not see a reason for a score change. The score would almost certainly remain at 8.

Reviewer LxMc was borderline (score 4) and focused on practicality: adjacency discovery, robustness to noisy adjacency, scalability/overhead, and comparison to retraining and newer baselines. The rebuttal made meaningful progress here. However, the reviewer explicitly followed up and stated they would maintain their original assessment.

Reviewer SCFy leaned mildly positive (score 6) but was uneasy about the theoretical components and asked for clearer explanations of PGD failure and the terms in Proposition 4.2. The rebuttal partially addressed these questions with clearer intuition, tighter explanations, and promised notation fixes. I would expect the score to stay the same.

Reviewer TtJK was borderline negative (score 4) and questioned the usefulness of the setting, the arbitrariness of defining "closely related" retain samples, missing intuition for the projection, outdated baselines, lack of retrain upper bound, and missing unlearning-time reporting. The rebuttal addressed most of these: it expanded the intuition for the projection, added newer baselines, fixed citations, reported runtime/memory, and added retrain comparisons. The core skepticism about defining adjacency in general settings likely remains, but several practical complaints were resolved. With full discussion, I would expect the score to move up slightly, from 4 to 6.

---

### Decision · Program_Chairs · 2026-01-26

Accept (Poster)